# Avoiding Undesired Future with Minimal Cost in Non-Stationary Environments

**Wen-Bo Du, Tian Qin, Tian-Zuo Wang, Zhi-Hua Zhou**
National Key Laboratory for Novel Software Technology, Nanjing University, China
School of Artificial Intelligence, Nanjing University, China
`{duwb, qint, wangtz, zhouzh}@lamda.nju.edu.cn`

## Abstract

Machine learning (ML) has achieved remarkable success in prediction tasks. In many real-world scenarios, rather than solely predicting an outcome using an ML model, the crucial concern is how to make decisions to prevent the occurrence of undesired outcomes, known as the *avoiding undesired future (AUF)* problem. To this end, a new framework called *rehearsal learning* has been proposed recently, which works effectively in stationary environments by leveraging the influence relations among variables. In real tasks, however, the environments are usually non-stationary, where the influence relations may be *dynamic*, leading to the failure of AUF by the existing method. In this paper, we introduce a novel sequential methodology that effectively updates the estimates of dynamic influence relations, which are crucial for rehearsal learning to prevent undesired outcomes in non-stationary environments. Meanwhile, we take the cost of decision actions into account and provide the formulation of AUF problem with minimal action cost under non-stationarity. We prove that in linear Gaussian cases, the problem can be transformed into the well-studied convex quadratically constrained quadratic program (QCQP). In this way, we establish the first polynomial-time rehearsal-based approach for addressing the AUF problem. Theoretical and experimental results validate the effectiveness and efficiency of our method under certain circumstances.

## 1 Introduction

Machine learning (ML) models have found extensive application in prediction tasks [25]. However, in contrast to a sole emphasis on prediction, it is preferred in many real-world scenarios to further explore effective decisions if the predicted outcomes are undesired. For instance, imagine that a factory manager has trained an ML model on features $\mathbf{X}$ (*e.g.*, economic indicators) to predict the outcome $\mathbf{Y}$ (*e.g.*, monthly sales). Suppose at the beginning of a month, $\mathbf{Y}$ is predicted to be undesired, *i.e.*, the predicted sales of the month are lower than expected. In this case, the manager usually wants to take action by altering some intermediate variables $\mathbf{Z}$ during the month to avoid this undesired outcome happening, *e.g.*, modifying the discount to attract more customers. The problem of how to find effective actions in such situations is known as avoiding undesired future (AUF) [63].

It is worth noting that AUF tasks often involve limited opportunities for interaction with the decision environment [63]. For instance, in the aforementioned example, the factory manager can only adjust the selling strategy once per month. Therefore, decision-making algorithms that depend on numerous interactions, such as conventional reinforcement learning (RL) methods [4], are not well-suited for the AUF problem [40]. Additionally, fundamentally vital decisions need to be made with human judgment, and therefore, it is desired to enable human decision-makers to understand why and how some actions can change the outcome. Due to these challenges, the structural relations among variables, which contain fine-grained information and are usually interpretable, are worth being considered to make decisions [36, 40].

38th Conference on Neural Information Processing Systems (NeurIPS 2024).

Causation is such a type of structural relation [36], which has been leveraged for some decision-making problems [5, 24, 42, 43]. Although causal relations can assist to some extent, identifying them is challenging and generally relies on some untestable assumptions [49, 44, 28]. Besides, causation should not be viewed as a prerequisite for decision-making problems, as humans can usually make good decisions without a thorough causal understanding [63]. Recognizing that *correlation* used

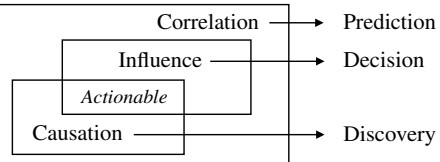

Figure 1: Relationship among correlation, influence, and causation [63].

in prediction is inadequate for decision-making whereas *causation* is too luxurious to be relied on, Zhou [63] emphasized the necessity of an intermediate relation that is stronger than correlation but less demanding than causation; this relation was subsequently called *influence relation* [64]. The relationship among *correlation*, *influence*, and *causation* is illustrated in Fig. 1.

Based on the influence relations, Qin et al. [40] developed the first *rehearsal learning* framework, which can effectively suggest good decision actions for the AUF problem in stationary environments. In practice, however, the method by Qin et al. [40] may lose its power when different decision actions are associated with different costs or when the decision environment is non-stationary where quantitative influence relations can vary. For example, modifying the discount into different levels results in different expenses, and the quantitative influence relation between pricing and sales can change seasonally for products such as the coat. Besides, it is worth mentioning that exact solutions are intractable with any polynomial-time algorithm when considering decisions on multiple variables in the previous work [40], thus exploring more efficient approaches is necessary for the decision-suggestion problem in the AUF problem.

To tackle these issues, in this paper, we propose the *AUF-MICNS* approach for the **AUF** problem with **mi**nimal **c**ost in **n**on-**s**tationary environments. AUF-MICNS considers a multi-round decision-making process where it suggests decisions and collects feedback data during and after each decision round. Note that although multiple decision rounds are allowed, the limited number of rounds may still render RL methods ineffective [40]. In contrast to Qin et al. [40], we consider the non-stationary fact that influence relations may vary over decision rounds, and therefore, treating the round-wise collected data as i.i.d. samples for determining the influence relations is inappropriate. To this end, we present a sequential approach to maintain dynamic influence relations in non-stationary environments, and further propose an online-ensemble-based [60] sequential algorithm to deal with the unknown degree of non-stationarity. In addition, we design a cost function to quantify the costs of different decision actions. The cost function takes into account not only (a) different unit costs associated with different variables, but also (b) the distinct costs involved in altering a single variable to varying extents. Further, we expect that the suggested decisions can efficiently avoid undesired outcomes with a relatively high probability. Rather than using the sampling-based method that cannot be solved with any polynomial-time algorithm [40], we reveal that finding decision suggestions for AUF with minimal action cost can be modeled as a convex quadratically constrained quadratic program (QCQP), which is solvable in polynomial time $\mathcal{O}(|\mathbf{V}|^3)$ with respect to the number of variables $|\mathbf{V}|$. Combining the parts above, we prove that our proposed approach can (a) accurately capture the dynamic influence relations with errors bounded by an exponentially decreasing term, and (b) efficiently suggest effective decision actions with minimal cost for the AUF problem.

Our main contributions are summarized as follows.

1. We try to tackle the AUF problem with minimal cost in non-stationary environments. Our modeling approach considers, for the first time, the decision action cost and the non-stationary fact that influence relations can vary over time in the AUF problem.

2. We present a novel sequential methodology to maintain dynamic influence relations. Theoretical results guarantee that the estimate error can be bounded by an exponentially decreasing term, as well as a fixed small value related to the problem difficulty.

3. We develop the AUF-MICNS algorithm, which is the first polynomial time rehearsal-based approach that can suggest effective decision actions for the formulated AUF problem. Our experimental results validate the effectiveness and efficiency of the method.

**Organization.** In Sec. 2, we review basic concepts and introduce our notation. In Sec. 3, we provide the formulation of AUF problem and provide the AUF-MICNS algorithm for solving the problem, together with theoretical guarantees. In Sec. 4, we introduce some related studies. In Sec. 5, we show the experimental results. At last, we discuss the limitations and conclude our work in Sec. 6.

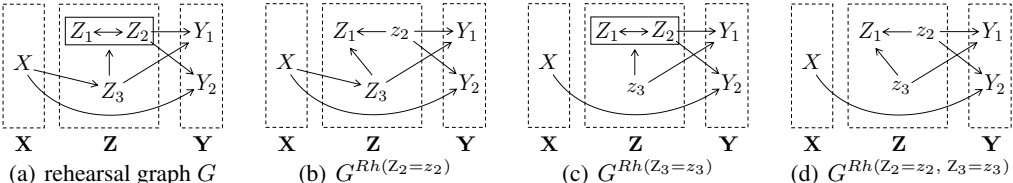

(a) rehearsal graph $G$     (b) $G^{Rh(Z_2=z_2)}$     (c) $G^{Rh(Z_3=z_3)}$     (d) $G^{Rh(Z_2=z_2,\ Z_3=z_3)}$

Figure 2: Fig. 2(a) is a rehearsal graph, and Fig. 2(b)∼ Fig. 2(d) illustrate the corresponding alteration graphs with different alterations. Note that when an alteration occurs to certain variables, all incoming arrows to those variables are removed while other graph structures are maintained.

## 2 Preliminaries

A novel probabilistic graphical model called structural rehearsal model (SRM) is proposed by Qin et al. [40] to characterize influence relations among variables in the AUF problem. The SRM consists of a set of rehearsal graphs and corresponding structural equations $\{\langle G_t, \boldsymbol{\theta}_t \rangle\}_t$ ($t$ denotes the decision round). The detailed definition of SRM is listed in Appendix B.1.

A rehearsal graph, denoted by $G = (\mathbf{V}, \mathbf{E})$, models the qualitative influence relations among variables. Specifically, the vertices $\mathbf{V}$ denote the variable set of the AUF problem and the edges $\mathbf{E}$ denote the influence relation among vertices. There are two types of edges in the rehearsal graph, the directional edge $X \to Y$ means that $X$ influences $Y$, and the bi-directional edge $X \leftrightarrow Y$ means that $X$ and $Y$ are mutually influenced. Additionally, the corresponding structural equations of variable $V_j$s can be parameterized by $\left\{ \beta_{j,t}, \sigma_{j,t}^2 \right\}_{j=1}^{|\mathbf{V}|} \subseteq \boldsymbol{\theta}_t$ that:

$$V_{j,t} := f_j \left( \mathrm{PA}_j^{G_t}; \beta_{j,t} \right) + \varepsilon_{j,t}, \tag{1}$$

where $V_j \in \mathbf{V}$ denotes the $j$-th vertex in $G_t$, $\mathrm{PA}_j^{G_t} \triangleq \{u \mid u \to V_j \text{ in } G_t\}$ represents the parents of $V_j$ in $G_t$, and the noise $\varepsilon_{j,t}$ follows the distribution $\mathcal{N}(0, \sigma_j{}^2)$ for all $t$. Note that on one hand, causal relations are not always necessary in real world decision-making problems [63]; on the other hand, causal models are sometimes insufficient for capturing the full scope of relationships between variables. For instance, the pressure and the temperature within a fixed volume of ideal gas are mutually influenced, as changes in either one of them affect the other. Such bi-directional influence relations are not well-represented by causal models, but can be naturally expressed by SRM [40].

Besides, finding suitable *alterations* is involved in addressing the AUF problem. An alteration $\xi$ means a decision action that is specified by human decision-makers, denoting by a set of vertex-value pairs, *e.g.*, $\xi = \{Z_2 = z_2\}$ in Fig. 2(b). Meanwhile, rehearsal operation, denoted as $Rh(\cdot)$, represents executing a certain alteration, which changes the original graph structure as illustrated in Fig. 2(b)-Fig. 2(d). Specifically, rehearsal operation breaks original influence links that point into any vertices contained in $\xi$, and fixes the values in $\xi$ to their associated vertices; while this operation maintains influence relations among other vertices in the associated alteration graph $G$. Since alteration and rehearsal operations are always considered together in the AUF problem, for simplicity, we will use the term *alteration* exclusively throughout the paper, unless otherwise specified.

## 3 The proposed approach

This section is dedicated to addressing the aforementioned AUF problem. In Sec. 3.1, we provide the probelm formulation. In Sec. 3.2, we propose the AUF-MICNS method for AUF. Later in Sec. 3.3, theoretical results are provided to ensure the effectiveness of our approach.

### 3.1 Formulation

This paper focuses on suggesting decisions to avoid undesired futures when an undesired outcome is predicted by an ML model. Since effective prediction models are widely applied in various domains [25] and work well even in non-stationary environments [62], we consider the case that the predictive ML model is always available and are not concerned about how to train it.

We formulate the AUF problem with minimal cost in non-stationary environments as a multi-round online decision-making process, where the decision-maker should perform round-wise alterations to

avoid undesired outcomes. In each decision round, there are two essential time points, the time that the ML prediction is made, and the time just before the generation of the concerned outcome $\mathbf{Y}$. As separated by the two time points, the variables fall into three consecutive time segments: $\mathbf{X}$, $\mathbf{Z}$, and $\mathbf{Y}$ as illustrated in Fig. 2(a). In $t$-th round, a decision-maker first observes variables $\mathbf{X}_t = \mathbf{x}_t$, and an ML model provides a prediction $\hat{\mathbf{Y}}_t$ as the outcome subsequently. Denote the desired region of $\mathbf{Y}_t$ as $\mathcal{S}$, if $\hat{\mathbf{Y}}_t \notin \mathcal{S}$, the decision-maker would perform alterations on $\mathbf{Z}_t$ (only once) based on the whole historical data, and the true $\mathbf{Y}_t$ will occur after the alteration.

It is worth mentioning that in decision round $t$, two crucial concerns for making decisions are (a) how to perform effective alterations that make $\mathbf{Y}_t \in \mathcal{S}$ with high probability; and (b) how to minimize the alteration cost as much as possible. For the former, we seek to ensure that $\mathbb{P}\left(\mathbf{Y}_t \in \mathcal{S} \mid \boldsymbol{\theta}_t, \mathbf{x}_t, Rh(\xi_t)\right) \geq \tau$; where $\xi_t$ is the selected alteration, and $\tau$ is the expected probability that the alteration can successfully avoid the undesired outcome. For the latter, we generalize a continuous cost function from the discrete cost measure [61] to quantify the cost for different alterations. The cost function is defined as the sum of the respective costs associated with each altered variable, and follows the increasing marginal cost property (see Appendix B.2 for details) in economics [32], properly measuring the alteration cost. Given the expectation that the future outcome will likely fall into $\mathcal{S}$ after performing the alteration, the decision maker consistently prefers the alteration with the least alteration cost. Thus, the AUF problem can be ideally formulated as follows in practice:

$$\min_{\xi_t} \quad \sum_{Z_i \in \xi_t} w_i \left(Z_i^{\xi_t} - Z_i^0\right)^2 \tag{2}$$
$$\text{s.t.} \quad \mathbb{P}\left(\mathbf{Y}_t \in \mathcal{S} \mid \boldsymbol{\vartheta}_t^\star, \mathbf{x}_t, Rh(\xi_t)\right) \geq \tau,$$

where $w_i > 0$ represents the cost coefficient for each intermediate variable $Z_i \in \mathbf{Z}$, $Z_i^{\xi_t}$ represents the values of $Z_i$ after alteration $\xi_t$ while $Z_i^0$ is the datum point that associates with the minimum alteration cost relatively. Besides, $\boldsymbol{\vartheta}_t^\star = \arg\min_{\boldsymbol{\vartheta}} \mathbb{E}_{\boldsymbol{\varepsilon}} \|\boldsymbol{\vartheta} - \boldsymbol{\theta}_t\|$ is the ideal estimation of $\boldsymbol{\theta}_t$ since $\boldsymbol{\theta}_t$ is not available in practice. Note that $w_i$s and $Z_i^0$s are user-specified because the cost of the same decision alteration can vary for different decision-makers.

In the following, we focus on a basic but essential class of the AUF problem, where the structural equations $f_t$ in Eq. (1) are linear but dynamic, the desired set $\mathcal{S}$ in Eq. (2) is a convex polytope, and the rehearsal graph $G$ is known and fixed (i.e., $G_t \triangleq G$) for a convenient illustration. Let $\mathbf{d} \in \mathbb{R}^s, \mathbf{M} \in \mathbb{R}^{s \times |\mathbf{Y}|}, \beta_j \in \mathbb{R}^{|\mathrm{PA}_j^G| \times 1}$, dynamic structural equations and the desired region can be formulated as:

$$V_{j,t} := \beta_{j,t}^T \mathrm{PA}_{j,t} + \varepsilon_{j,t}, \ \mathcal{S} = \left\{\mathbf{y} \in \mathbb{R}^{|\mathbf{Y}|} \mid \mathbf{My} \leq \mathbf{d}\right\}. \tag{3}$$

Note that $\boldsymbol{\vartheta}_t^\star$ in Eq. (2) is usually not available as discussed later in Sec. 3.2.1, thus finding surrogate estimations $\hat{\boldsymbol{\theta}}_t$ with bounded error $\mathbb{E}_{\boldsymbol{\varepsilon}} \|\hat{\boldsymbol{\theta}}_t - \boldsymbol{\theta}_t\|$ is necessary. Noticing that the noise distributions for $\varepsilon_{j,t}$s in Eq. (3) do not change over time, the variances of $\varepsilon_{j,t}$s can be estimated by various Bayesian learning methods [6]. Thus, we mainly focus on estimating influence parameters $\beta_{j,t}$s in what follows, and we use $\boldsymbol{\theta}_t$ to represent $\{\beta_{j,t}\}$s only unless otherwise specified. Another challenge lies in how to efficiently solve the optimization Eq. (2) under the probabilistic constraint, since the sampling method used in the previous work has been proven to be time-consuming [40]. In Sec. 3.2, we propose the AUF-MICNS algorithm to tackle the aforementioned two challenges. AUF-MICNS can maintain the dynamic influence relations accurately and suggest alterations to avoid undesired outcomes effectively, and the performance can be guaranteed by theoretical results in Sec. 3.3.

### 3.2 Our proposed AUF-MICNS algorithm

In this subsection, we propose the AUF-MICNS algorithm to address the AUF problem as formulated in Eq. (2). AUF-MICNS consists of two components, influence maintenance and alteration suggestion. We first introduce the components respectively, then we discuss how to combine them together.

#### 3.2.1 Dynamic influence maintenance

As illustrated in Eq. (2), the precondition for making effective decisions in the AUF problem lies in the accurately estimated $\boldsymbol{\theta}_t$s. Specifically, to accurately estimate the parameter vector $\beta_{j,t} \in \boldsymbol{\theta}_t$ in $t$-th decision round, we would like to minimize the expected squared loss $\mathbb{E}_{\varepsilon_j}\left[V_j - \beta_{j,t}^\top \mathrm{PA}_j\right]^2$,

where $V_j$ is the $j$-th vertex in rehearsal graph, and $\mathrm{PA}_j$ denotes the parents of $V_j$. Ideally, according to the law of large numbers (LLN), if we could obtain sufficient i.i.d. samples in time $t$, we can estimate $\beta_{j,t}$ by minimizing the empirical error $\ell_{j,t}(\beta_{j,t})$ as a substitute:

$$\underset{\beta_{j,t}}{\arg\min} \ \frac{1}{2n} \sum_{k=1}^{n} \left( V_{j,t}^k - \beta_{j,t}^\top \mathrm{PA}_{j,t}^k \right)^2, \tag{4}$$

where $n$ is the number of samples that we can obtain in round $t$, and the superscript $k$ means that it is the $k$-th sample. It can be guaranteed in lemma C.3 that this estimation converges to the true value as $n$ increases. However, in real-world cases, we can only obtain a single sample after each decision round, so it is inappropriate to calculate the parameters as above, since the solution to Eq. (4) is far from accurate in this data-limited situation, where the surrogate loss is:

$$\hat{\ell}_{j,t}(\beta_{j,t}) = \frac{1}{2} \left( V_{j,t} - \beta_{j,t}^\top \mathrm{PA}_{j,t} \right)^2. \tag{5}$$

Besides, mixing the round-wise selected $\{V_{j,t}, \mathrm{PA}_{j,t}\}_{t=1}^T$ to estimate parameter $\beta_{j,t}$ as the classic empirical risk minimization (ERM) [34] methods do is inappropriate as well, because data selected in different rounds possibly obeys different distributions in non-stationary environments. Fortunately, in the field of online learning [52, 14, 17], there exist many types of algorithms to estimate parameters sequentially with limited samples. Online gradient descent (OGD) [65, 47] is a typical class of online learning algorithms, which takes advantage of the gradient descent idea to handle the round-wise selected data. Once the selection of the step size (learning rate) in OGD is tailored to the varying speed of the environment, *i.e.*, employing a larger step size for rapid changes and a smaller one for gradual changes, parameters can be effectively updated with limited data after each round. We customize OGD for estimating quantitative influence relations $\hat{\beta}_{j,t} \in \boldsymbol{\theta}_t$ sequentially in Algorithm 1.

By using the OGD-based approach in Algorithm 1, we can obtain a sequence of estimates $\{\hat{\beta}_{j,1}, \ldots, \hat{\beta}_{j,T}\}$ for $j \in [|\mathbf{V}|]$. Roughly speaking, the quality of the estimates heavily depends on the choice of step size $\eta$. If we have full prior knowledge of the non-stationarity degree, such as the changing speed of the influence relations, we can pre-determine an optimal step size $\eta^\star$ to achieve favorable estimates. However, in practical scenarios, $\eta^\star$ is not available, and the random choice of the step size leads to unstable estimators.

---

**Algorithm 1** OGD-based estimator for $\hat{\beta}_j$

**Input:** The step size $\eta$

1: Initialize $\hat{\beta}_{j,0}^\eta$ with any point in domain $\mathcal{B}$
2: **for** $t = 1$ **to** $T$ **do**
3:     Receive $(\mathrm{PA}_{j,t}, V_{j,t})$; **Continue** if $V_{j,t} \in \xi_t$
4:     Estimate gradient $\hat{g}_{j,t} = \left( \mathrm{PA}_{j,t}^\top \hat{\beta}_{j,t}^\eta - V_{j,t} \right) \mathrm{PA}_{j,t}$
5:     Update $\hat{\beta}_{j,t+1}^\eta = \Pi_{\mathcal{B}} \left[ \hat{\beta}_{j,t}^\eta - \eta \hat{g}_{j,t} \right]$

**Output:** estimated $\{\hat{\beta}_{j,t}^\eta\}_{t=1}^T$

---

Since bad estimators of the influence relations will affect the accuracy of the estimated distribution $\mathbb{P}(\mathbf{Y}_t \in \mathcal{S} \mid \hat{\boldsymbol{\theta}}_t, \mathbf{x}_t, Rh(\xi_t))$, which will further affect the effectiveness of the suggested decisions for the AUF problem, thus the choice of the step size needs to be carefully considered.

To avoid this risk, we turn to use online-ensemble [60] based methods. Online ensemble is a type of algorithm that combines the idea of ensemble and sequential updating, maintaining multiple base learners and ensembling them together. Specifically, as illustrated in Algorithm 2, we maintain $N_j$ base estimators with a weight vector $w$. After each round, all experts update their estimates by the collected data, and the weight vector $w$ will be updated as well by different losses related to different experts. By ensembling all experts with the weight $w$, we can obtain the final estimator.

---

**Algorithm 2** Online-ensemble-based estimator for $\hat{\beta}_j$

**Input:** base estimators' number $N_j$, weight parameter $\alpha$

1: Set a set of learning rates $\mathcal{H}_j = \left\{ \eta^i \mid i = 1, \ldots, N_j \right\}$
2: Initialize weight vector $w_t^\eta = \frac{1}{N_j}$ for each $\eta \in \mathcal{H}_j$
3: Activate estimators $E^\eta$s for all $\eta \in \mathcal{H}_j$ by OGD with $\eta$
4: **for** $t = 1$ **to** $T$ **do**
5:     Receive $\beta_{j,t}^\eta$ from each $E^\eta$
6:     Output $\hat{\beta}_{j,t}$ as $\sum_{\mu \in \mathcal{H}_j} w_t^\mu \beta_{j,t}^\mu$
7:     Update weights as
8:     $w_{t+1}^\eta = \frac{w_t^\eta \exp\left(-\alpha \hat{\ell}_{j,t}(\beta_{j,t}^\eta)\right)}{\sum_{\mu \in \mathcal{H}_j} w_t^\mu \exp\left(-\alpha \hat{\ell}_{j,t}(\beta_{j,t}^\mu)\right)}$
9:     Receive $(\mathrm{PA}_{j,t}, V_{j,t})$ and send to each $E^\eta$

**Output:** estimated $\{\hat{\beta}_{j,t}\}_{t=1}^T$

---

### 3.2.2 Efficient alteration suggestion

In $t$-th decision round, since we can obtain the estimation $\hat{\boldsymbol{\theta}}_t$ for true $\boldsymbol{\theta}_t$ by Algorithm 2, then we substitute $\boldsymbol{\vartheta}^\star$ with $\hat{\boldsymbol{\theta}}_t$ in Eq. (2). To suggest alterations efficiently and effectively, another crucial aspect is finding ways to delineate the feasible domain of the probabilistic constraint in Eq. (2). We first present a supporting result in Lemma 3.1 as follows, which is needed in our method to find the alteration that can make $\mathbf{Y}_t \in \mathcal{S}$ with an expected probability $\tau$.

**Lemma 3.1** (Qin et al., 2023). *Given* $\mathbf{x}_t, \boldsymbol{\theta}_t$, *it holds that:*

$$\mathbf{Y}_t = \mathbf{A}\mathbf{x}_t + \mathbf{B}\mathbf{z}_t^\xi + \mathbf{C}\boldsymbol{\varepsilon}_t,$$

*where* $\mathbf{A}, \mathbf{B}, \mathbf{C}$ *are constant matrices of appropriate shapes based on* $\boldsymbol{\theta}_t$, $\boldsymbol{\varepsilon}_t = [\varepsilon_{1,t}, \dots, \varepsilon_{|\mathbf{V}|,t}] \sim \mathcal{N}(\mathbf{0}, \boldsymbol{\Sigma})$, *and* $\mathbf{z}_t^\xi$ *are intermediate variables with alteration* $\xi$.

Recall from Eq. (2) that we want to find alterations that satisfy $\mathbb{P}(\mathbf{Y}_t \in \mathcal{P} \mid \boldsymbol{\theta}_t, \mathbf{x}_t, Rh(\xi_t)) \geq \tau$. Recognizing that solving optimization with probabilistic constraint is generally intractable and the previous sampling method is time-consuming, we attempt to construct a surrogate deterministic constraint that can be handled efficiently to replace the probabilistic constraint. Fortunately, this idea is feasible because Lemma 3.1 shows that once the alteration is selected, the randomness of $\mathbf{Y}_t$ only arises from $\boldsymbol{\varepsilon}_t$, since $\mathbf{x}_t$ has been observed and $\mathbf{A}, \mathbf{B}, \mathbf{C}$ are constant matrices given $\boldsymbol{\theta}_t$. Thus, the probability density function (PDF) of $\mathbf{Y}_t$ is available and we can directly analyze the PDF to find probability regions as in Prop. 3.2, which aids in constructing the surrogate deterministic constraint.

**Proposition 3.2.** *The following probability region* $\mathcal{P}$ *satisfies* $\mathbb{P}(\mathbf{Y}_t \in \mathcal{P} \mid \boldsymbol{\theta}_t, \mathbf{x}_t, Rh(\xi_t)) = \tau$:

$$\mathcal{P} = \left\{ \boldsymbol{\mu}_{\mathbf{y}_t} + \left( \chi^{-1}(\tau)\mathbf{C}\boldsymbol{\Sigma}\mathbf{C}^\top \right)^{\frac{1}{2}} \mathbf{u} \;\middle|\; \|\mathbf{u}\|_2 \leq 1 \right\},$$

*where* $\boldsymbol{\mu}_{\mathbf{y}_t} = \mathbf{A}\mathbf{x}_t + \mathbf{B}\mathbf{z}_t^\xi$, $\mathbf{u}$ *is an arbitrary point in the unit sphere in* $\mathbb{R}^{|\mathbf{Y}_t|}$, *and* $\chi^{-1}(\cdot)$ *denotes the quantile function of the* $\chi^2$ *distribution with degrees of freedom* $\lambda = |\mathbf{Y}_t|$.

The proof of Prop. 3.2 is provided in Appendix C.4. We raise the power of $1/2$ to the matrix because it is always positive semi-definite since $\chi^{-1}(\cdot) \geq 0$ and $\mathbf{C}\boldsymbol{\Sigma}\mathbf{C}^\top \succeq 0$ ($\boldsymbol{\Sigma}$ is the covariance matrix). In practical scenarios where $\boldsymbol{\Sigma}$ is not available, the estimation $\hat{\boldsymbol{\Sigma}}$ can be used as replacements. Let $F_{\mathbf{y}_t}$ denote the cumulative distribution function (CDF) of $\mathbf{Y}_t$, Fig. 3 illustrates a 2-dimensional example of samples from $F_{\mathbf{y}_t}$, together with the estimation of the associated region $\hat{\mathcal{P}}$. It illustrates that $\hat{\mathcal{P}}$ can properly draw the probabilistic region of $F_{\mathbf{y}_t}$ with expected probability $\tau$.

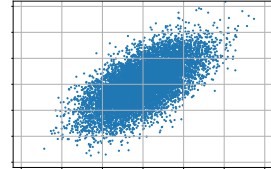 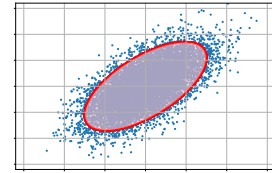

(a) Samples from $F_{\mathbf{y}_t}$      (b) Probability region $\hat{\mathcal{P}}$

Figure 3: An example of estimated $\hat{\mathcal{P}}$ with $\tau = 0.9$

Moreover, recognizing that the desired region $\mathcal{S}$ defined in Eq. (3) is a convex polytope, *i.e.*, $\mathcal{S} = \left\{ \mathbf{y} \in \mathbb{R}^{|\mathbf{Y}|} \mid \mathbf{M}\mathbf{y} \leq \mathbf{d} \right\}$, we utilize the defined probability region $\mathcal{P}$ in Prop. 3.2, and use a deterministic constraint to replace the original probabilistic constraint in Eq. (2). In this case, the alteration-suggestion method can be formulated as follows:

$$\min_{\mathbf{z}_t^{\xi_t}} \quad \left( \mathbf{z}_t^{\xi_t} - \mathbf{z}_t^0 \right)^\top \mathbf{W} \left( \mathbf{z}_t^{\xi_t} - \mathbf{z}_t^0 \right)$$

$$\text{s.t.} \quad \mathbf{M}\mathbf{A}\mathbf{x}_t + \mathbf{M}\mathbf{B}\mathbf{z}_t^{\xi_t} + \|\mathbf{M}\mathbf{P}\|_{2,\text{row}} \leq \mathbf{d}, \tag{6}$$

where $\mathbf{P} = \left( \chi^{-1}(\tau)\mathbf{C}\boldsymbol{\Sigma}\mathbf{C}^\top \right)^{\frac{1}{2}}$, $\|\cdot\|_{2,\text{row}}$ means an operator that takes 2-norm for each row of the matrix thus outputs a row-dimensional vector. The objective to be minimized is the vector representation of the cost function as explained in Eq. (2), where $\mathbf{W} = \text{diag}(w_1, \dots, w_{|\mathbf{z}_t^{\xi_t}|})$ is positive definite since the cost is non-negative, *i.e.*, $w_i > 0$ for $\forall i$.

Note that $\mathbf{W}$ is positive definite and the constraint is linearly associated with $\mathbf{z}_t^{\xi_t}$, thus Eq. (6) is a convex QCQP, which can be cast as a second-order cone program [35, 30]. Specifically, it can be solved in polynomial time $\mathcal{O}(|\mathbf{Z}_t^{\xi_t}|^3 \cdot L)$ by interior-point method [2], where $L$ is the iteration rounds

for solving the QCQP and is not associated with $|\mathbf{Z}_t^{\xi_t}|$. Meanwhile, constructing matrices $\mathbf{A}, \mathbf{B}, \mathbf{C}$ needs an element-wise traverse thus $\mathcal{O}(|\mathbf{V}|^2)$ time and constructing the probability region $\mathbf{P}$ as in Prop. 3.2 runs under $\mathcal{O}(|\mathbf{V}||\mathbf{Y}|^2)$ because of the matrix multiplication, so the whole running time of alteration suggestion is $\mathcal{O}(|\mathbf{V}|^3)$ as $|\mathbf{Z}_t^{\xi_t}| = \Theta(|\mathbf{V}|), |\mathbf{Y}| = \Theta(|\mathbf{V}|)$ and $L$ is a constant.

### 3.2.3 AUF-MICNS

By combining the influence mainte- nance step and the alteration sugges- tion step, our proposed approach for addressing the AUF problem with minimal cost in non-stationary en- vironments is formulated in Algo- rithm 3, which attempts to avoid unde- sired outcomes in each decision round. Specifically, in $t$-th round, the algo- rithm first receive $\mathbf{X}_t = \mathbf{x}_t$, then if the predicted outcome $\hat{\mathbf{Y}}_t \notin \mathcal{S}$, the

---

**Algorithm 3** AUF-MICNS

**Input:** sequential coming data $\{\mathbf{x}_t, \mathbf{z}_t, \mathbf{y}_t\}_{t=1}^T$

1: Initialize $\{\hat{\beta}_{j,0}\}_{j=1}^{|\mathbf{V}|}$ for Algorithm 2
2: **for** $t = 1$ **to** $T$ **do**
3:     Select alteration $\xi_t$ by solving Eq. (6)
4:     Receive $\mathbf{y}_t$ and sent $\{\mathbf{x}_t, \mathbf{z}_t^{\xi_t}, \mathbf{y}_t\}$ to Algorithm 2
5:     Update $\{\hat{\beta}_{j,t}\}_{j=1}^{|\mathbf{V}|}$ by Algorithm 2

**Output:** suggested alterations $\{\xi_t\}_{t=1}^T$

---

algorithm performs the suggested alteration $\xi_t$ on $\mathbf{Z}_t$ by solving Eq. (6). Subsequently, true $\mathbf{Y}_t = \mathbf{y}_t$ occurs, and the algorithm collects $\{\mathbf{x}_t, \mathbf{z}_t^{\xi_t}, \mathbf{y}_t\}$ to update influence relations by Algorithm 2. By using this algorithm, one can tackle the formulated AUF problem with the suggested alterations.

## 3.3 Theoretical results

In this subsection, we present the theoretical analysis of our proposed method. All proofs are given in Appendix C. First, we can determine the dynamic influence relations in non-stationary environments with theoretical guarantees. Specifically, by using Algorithm 1 to estimate $\beta_{j,t} \in \boldsymbol{\theta}_t$, the error gap between the estimate and the true parameter value ($\mathbb{E}_{\varepsilon_j}\|\hat{\beta}_{j,t} - \beta_{j,t}\|^2$) is proved to be bounded by an exponentially decreasing term, as well as a fixed value related to the choice of step size and the inherent problem difficulty. It reveals that the performance of Algorithm 1 depends on the choice of step size $\eta$ heavily, and is detailed in Thm. 3.3, where $\{\ell_{j,t}(\cdot)\}_{t=1}^T$s are defined in Eq. (4).

**Theorem 3.3.** *Let $\beta_{j,t}$ ($j \in [|\mathbf{V}|]$) denote the true parameter value of $\beta_j$ in time $t$, and choose $\eta_j \in (0, 1/2L_j]$ as the step size used in Algorithm 1, then it can be bounded that:*

$$\mathbb{E}\left\|\hat{\beta}_{j,t} - \beta_{j,t}\right\|^2 \lesssim (1 - \mu_j\eta_j)^{\frac{t}{m}}\left\|\hat{\beta}_{j,0} - \beta_{j,0}\right\|^2 + \delta_j \ \ with \ \ \delta_j = \left(\frac{m\Delta_j}{\mu_j\eta_j}\right)^2 + \frac{\eta_j\sigma^2}{\mu_j},$$

*where $\mu_j$ and $L_j$ are the minimal and maximal eigenvalues of $\{\ell_{j,t}(\cdot)\}_{t=1}^T$'s Hessian matrices, $\sigma^2$ upper-bounds the variance of $\hat{g}_{j,t}$, $\Delta_j \geq \max_t \|\beta_{j,t+1} - \beta_{j,t}\|$ upper-bounds the varying speed of the environment, and $m$ is the longest continuously altered rounds of $V_j$, for most of the $V_j$s, $m = 1$.*

In Thm. 3.3, it holds that $\mu_j\eta_j \in (0, 1/2]$, which derives that $(1 - \mu_j\eta_j) \in [1/2, 1)$. This shows that as time progresses, the OGD estimator $\hat{\beta}_{j,t}$ will gradually converge towards the true value $\beta_{j,t}$ in the expected sense. Specifically, (a) the convergence speed depends on the choice of the initial point $\hat{\beta}_{j,0}$, and is limited by the inherent difficulty of the problem implied in $\mu_j/L_j$; and (b) the convergence result will suffer a $\delta_j$ gap from the true value, depending on the varying speed of the environment ($\Delta_j$) and the choice of step size $\eta_j$. If $\eta_j$ is appropriately chosen, the gap $\delta_j$ will be small, *e.g.*, if all hyperparameters are available and $\Delta_j \neq 0$, choosing $\eta_j^\star = \min\{1/2L_j, \sqrt[3]{2}(\frac{m\Delta_j}{\sqrt{\mu_j}\sigma})^{\frac{2}{3}}\}$ can achieve the smallest $\delta_j$. Meanwhile, the hyperparameter $m$ appears in the bound according to the properties of alterations shown in Fig. 2. If $V_j$ is continuously altered in $m$ rounds, all incoming arrows of $V_j$ will be removed and the parameters associated with the arrows will not be updated in those rounds. Because only a part of vertices in $\mathbf{Z}_t$ might be altered in any rounds, $m = 1$ holds for most of the $V_j$s.

In practice, we do not know exact parameters such as $\Delta_j$, thus $\eta^\star$ is not pre-available. Recognizing that the random choice of $\eta$ leads to unstable estimations, we turn to use Algorithm 2 to estimate $\hat{\beta}_{j,t}$s. Though we do not know $\eta^\star$ as well, by using the online-ensemble-based Algorithm 2, we can get more stable estimations as guaranteed by the following regret bound in Prop. 3.4.

**Proposition 3.4.** *Assume* $\{\hat{\ell}_{j,t}(\cdot)\}_{t=1}^T$*s are bounded for* $\forall \beta_i \in \mathcal{B}$ *and* $t \in [T]$*; then for any* $\eta \in \mathcal{H}_j$*, estimations* $\hat{\beta}_{j,t}$*s from Algorithm 2 satisfies that*

$$\sum_{t=1}^T \hat{\ell}_t \left( \hat{\beta}_{j,t} \right) - \sum_{t=1}^T \hat{\ell}_t \left( \beta_{j,t}^\eta \right) \leq \mathcal{O} \left( \sqrt{T \ln N_j} \right);$$

*by choosing* $\alpha = \sqrt{\ln N_j / T}$ *in Algorithm 2, where* $N_j$ *is the number of base-learners,* $\beta_{j,t}^\eta$ *is the estimation from any expert* $\eta$ *in expert set* $\mathcal{H}_j$ *in Algorithm 2.*

Prop. 3.4 shows that the cumulative loss of the estimation obtained by Algorithm 2 is comparable with the best expert in $\mathcal{H}_j$, thus by Thm. 3.3, though $\hat{\boldsymbol{\theta}}_t$ may be far from $\boldsymbol{\theta}_t$ at the first few rounds, it will converge towards $\boldsymbol{\theta}_t$. Meanwhile, in some certain cases, the best step size $\eta_j^\star$ in Thm. 3.3 can be included in $\mathcal{H}_j$ with $N_j = \mathcal{O}(\log T)$ [60]. Due to these, Algorithm 2 provides more stable estimates for the influence relations, which can further aid in making decisions to address the formulated AUF problem. As to the decision-making processes, AUF-MICNS can suggest alterations with theoretical guarantees. Specifically, the suggested alterations are guaranteed by Thm. 3.5.

**Theorem 3.5.** *By using the suggested alterations* $\xi_t$ *from Eq. (6), it can be guaranteed that:*

$$\mathbb{P} \left( \mathbf{Y}_t \in \mathcal{S} \mid \hat{\boldsymbol{\theta}}_t, \mathbf{x}_t, Rh(\xi_t) \right) \geq \tau.$$

Thm. 3.5 illustrates that the suggested alterations $\xi_t$ by the AUF-MICNS algorithm can effectively avoid the undesired outcomes as the formulated AUF problem expects, under the distribution conditioned on estimation $\hat{\boldsymbol{\theta}}_t$ rather than true $\boldsymbol{\theta}_t$. Note that it is guaranteed in Thm. 3.3 and Prop. 3.4 that $\hat{\boldsymbol{\theta}}_t$ is not far from $\boldsymbol{\theta}_t$. Meanwhile, we provide the experimental analysis in Sec. 5 to show that $\mathbb{P} \left( \mathbf{Y}_t \in \mathcal{S} \mid \boldsymbol{\theta}_t, \mathbf{x}_t, Rh(\xi_t) \right) \geq \tau$ holds practically. Besides, this method achieves a super-exponential improvement over the time complexity of the previous method [40] as discussed in Sec. 3.2.2.

## 4   Related work

**RL Methods.** RL approaches have demonstrated success in numerous domains [51], particularly in game-playing [33] and autonomous control [21]. However, the Markov Decision Process (MDP) formalism in RL abstracts decision-making processes into states, actions, and rewards, potentially overlooking useful fine-grained structural information. While hybrid online and offline RL methods have been introduced [48, 37], they overlook fine-grained structural information as well and require large offline datasets in practice. Moreover, it is worth noting that a fundamental limitation of applying RL to the AUF problem is that RL methods require a substantial number of interactions, which may be too luxurious or simply not tolerated in many real-world applications.

**Causality.** Identifying causal systems from observational and interventional data has been extensively studied [44, 50, 7, 58, 9], but these methods typically do not actively select interventions. Furthermore, significant research has focused on identifying causal structures or effects in interactive environments [18, 22, 53, 55, 39, 54, 57, 41], which predominantly aim to identify causal structures or effects. To incorporate additional utilities for decision-making, causal bandits and causal RL methods have been proposed to determine where to intervene [5, 24, 46, 26, 59, 12, 31]. The aforementioned methods generally rely on causal modeling, which may be luxurious or restrictive in some real-world decision-making cases [63, 40]. As identifying causations rely on some strong restrictions or assumptions, it is possible that we cannot find a feasible alteration. Conversely, correlation, which underpins most ML models, falls short of providing a solid foundation for making decisions. As a middle ground between causation and correlation, the influence relations form a more practical basis for decision-making [63]. Building on this concept, we employ the SRM developed by Qin et al. [40], which can adapt to dynamically evolving decision systems. In particular, our approach incorporates the context $\mathbf{x}_t$ into decision-making, and can sugggest decision alterations effectively even when the parameters of the system are non-stationary and the actions come with varying associated costs.

**Other related topics.** Our approach builds upon several classic ML techniques. The action cost measure employed in our method generalizes principles from cost-sensitive learning [13, 61]. Additionally, we adapt the online ensemble methodology [60] to update the SRM parameters. Online emsemble framework has been used in several areas, such as online convex optimization [56], online label shift [38], and reinforcement learning [27]. Further exploration and advancements in these techniques hold the potential to enhance our approach as well.

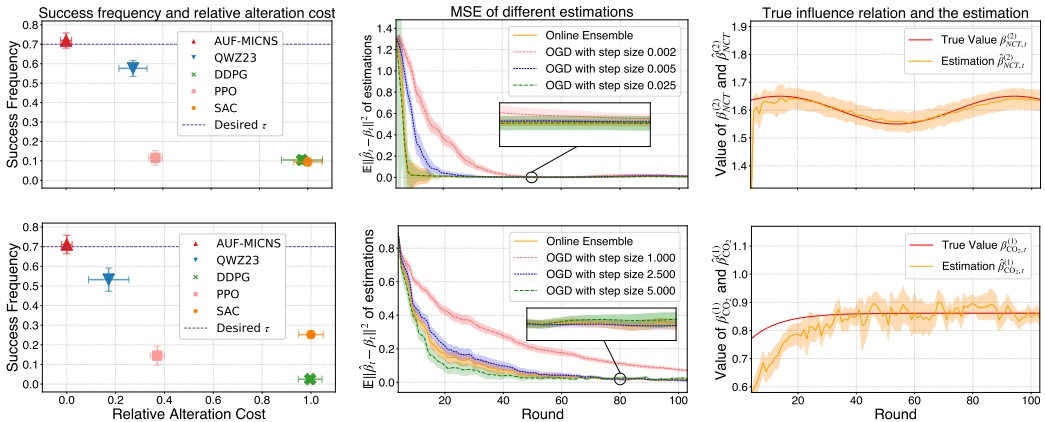

Figure 4: Results of Market-managing data (row 1) and Bermuda data (row 2) respectively. Bars and bands depict the standard deviations.

## 5 Experiments

We evaluate the proposed AUF-MICNS algorithm on two datasets and focus on four aspects including (a) success frequency for $\mathbf{Y}_t \in \mathcal{S}$; (b) average alteration cost; (c) average executing time; and (d) mean square error of the estimated parameters $\hat{\boldsymbol{\theta}}_t$. For each dataset, we repeat experiments with 100 rounds 20 times. The observational dataset size is set to 10. We compare our proposed method with previous rehearsal learning approach (QWZ23 [40]), and we also compare with several classic RL methods including DDPG [29], PPO [45], and SAC [15]. Experimental details and additional experiments are listed in Appendix D.

**Market-Managing Data.** We abstract a dynamic SRM from a market-managing scenario, where a manager of the market needs to make decisions to promote the total profit (TPF) and the number of customers (NCT). We consider variables that may affect TPF and NCT, the pricing for the product (P), the pricing of the competitor market (E), the cost of raw materials (C), etc. We assume the manager can alter two variables, P and C. There exist mutually influenced variables in the scenario: If P is set to be small, then E will also be small to stay competitive on price; and vice versa. The sizes of $\mathbf{X}_t$, $\mathbf{Z}_t$, and $\mathbf{Y}_t$ are 2, 4, and 2 respectively. The parameters of the dynamic structural equations are manually set according to the domain knowledge. For example, parameters associated with variable C vary over time with a periodic term, *i.e.*, $\beta_{C,t} = \bar{\beta}_C(1 + a\sin(wt))$. The feasible alteration values are $[-3, 3]$ for centralized P and C, associated with cost coefficients $1.0$ and $2.0$ respectively since altering P is easier than C. We want to maintain high TPF and high NCT at the same time, so the desired region $\mathcal{S}$ is set to be $\mathcal{S} = \{\text{TPF} \geq s_1, \text{NCT} \geq s_2, \text{TPF} + \text{NCT} \geq s_3\}$, and more than 80% of the original data falls outside this range, as shown in Fig. 7(a).

**Bermuda Data.** This is an ecology dataset that records environment variables in Bermuda [10], and the generation order of variables is available [3]. The sizes of $\mathbf{X}_t$, $\mathbf{Z}_t$, and $\mathbf{Y}_t$ are 3, 7, and 1. The structural equations are obtained by fitting linear models on normalized data [40], and we manually add the varying trend, *e.g.*, considering the annual increase in $CO_2$ emissions, we posit that there is an increasing trend in the influence relation between temperature and $CO_2$ concentration, *i.e.*, $\beta_{CO_2,t} = \bar{\beta}_{CO_2,t} \cdot (1 - a\,e^{-wt})$. We assume that 5 variables in $\mathbf{Z}_t$ are actionable [1] and the feasible alteration values are $[-1, 1]$ for each of them with different cost coefficients. The concerned outcome $\mathbf{Y}_t$ represents the net coral ecosystem calcification (NEC) in Bermuda. To make the coral reef ecosystem healthy, a relatively large NEC is preferred, so the desired region $\mathcal{S}$ is set to be $\mathcal{S} = \{0.5 \leq \text{NEC} \leq 2\}$, and more than 75% of the original data falls outside it, as shown in Fig. 7(b).

Fig. 4 shows the full experimental results. The desired probability is set to $\tau = 0.7$. Two rows of figures denote the results of the Market data and the Bermuda data respectively. Specifically, the first column of Fig. 4 shows that (a) the success frequency of making the concerned outcome $\mathbf{Y}_t$ falls into the desired region $\mathcal{S}$; and (b) relative alteration cost among methods, *i.e.*, the normalized cost among different methods. The proposed AUF-MICNS algorithm outperforms other competitors in both aspects. If we increase the interaction rounds $T$, RL methods can achieve satisfying performance, *e.g.*, DDPG achieves 0.6955 average success frequency when $T = 4000$ on the Market data.

The second column of Fig. 4 shows the mean-square error of estimates for $\beta_{\text{NCT}}$ in Market data and estimates for $\beta_{\text{CO}_2}$ in Bermuda data respectively. The exponential convergence speed and the convergence error ($\delta_j$) in Thm. 3.3 are illustrated in these two figures. Specifically, for Market-managing data, the online-ensemble-based sequential method can achieve a similar convergence speed like OGD with $\eta = 0.025$ does when $T \leq 20$, and it results in a smaller $\delta_j$ as illustrated by the enlarged part ($T \geq 40$). For Bermuda data, OGD with $\eta = 2.5$ outperforms other step sizes, since $\eta = 1.0$ converge slowly, and $\eta = 5.0$ suffers a notably bigger gap $\delta_j$. By using the online-ensemble-based sequential method, we can obtain a comparable performance with the best step size $\eta = 2.5$. These results illustrate that our proposed method mitigates the risk substantially, as the inappropriate choice of $\eta$ will affect the practicality of the estimation and $\eta^\star$ is not available practically. Meanwhile, the third column of Fig. 4 shows true parameter values and the estimates by the online-ensemble-based method. The superscript $(1)/(2)$ means the first/second dimension of the associated parameter vector. It shows that the error gap between $\hat{\beta}_{j,t}$ and $\beta_{j,t}$ converges to a small value rapidly, which guarantees accurate estimates of the quantitative influence relations in possibly non-stationary environments.

At last, the average whole-executing time of the 20-times experiments is recorded in Table 1. We mainly focus on the comparison between AUF-MICNS and QWZ23 since both of them maintain influence relations other than purely suggesting decisions.

Table 1: Average running time (s).

| Dataset | DDPG | PPO | SAC | QWZ23 | MICNS |
|---------|------|-----|-----|-------|-------|
| Market  | 7.89 | 0.05 | 0.03 | 63.14 | 2.81 |
| Bermuda | 9.63 | 0.06 | 0.04 | 386.44 | 1.71 |

It shows that our proposed method is more time-efficient than QWZ23.

## 6 Conclusion

Practically, different decision actions might correspond to different costs, and the influence relations might vary over time in non-stationary environments. In this paper, we try to tackle the AUF problem considering the aforementioned aspects. Specifically, we propose the AUF-MICNS algorithm that can capture the dynamic influence relations in non-stationary environments and suggest actions based on the influence relations. This method can suggest decision actions under polynomial time. Meanwhile, theoretical results show that the suggested actions can effectively avoid the undesired outcomes with probability larger than $\tau$, and the suggested actions get more accurate as time progresses. Experimental results validate the effectiveness and efficiency of our proposed method.

Our approach primarily focuses on scenarios where the influence relations among variables are linear and is currently not applicable to non-linear cases. Additionally, selecting the hyperparameter $\tau$ in existing frameworks remains a significant challenge, as an inappropriate $\tau$ may lead to the failure of solving Eq. (6). To address these limitations, we plan to develop methods that handle non-linear cases and reduce sensitivity to the hyperparameter $\tau$ in future work.

## Acknowledgements

This research was supported by Jiangsu Science Foundation Leading-edge Technology Program (BK20232003), and National Postdoctoral Program for Innovative Talent (BX20240162). The authors thank the anonymous reviewers for their helpful comments. We are also grateful to Long-Fei Li and Lue Tao for discussions.

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

# A Discussion

## A.1 Discussion on the AUF-MICNS algorithm

Two crucial points should be discussed in this approach, *i.e.*, (a) dynamic influence modeling; and (b) the defined cost function for alterations. For the former, we consider a linear case with additive noise, and for more complicated scenarios than linear ones, one can modify the structural equations in Eq. (3) by using methods such as the kernel method [20]. Note that Theorem 3.3 always holds as long as the new loss functions $\ell_{j,t}(\cdot)$s are still $\mu$-strongly convex and $L$-smooth in those cases, similar to Eq. (4). For the latter, we use a quadratic cost function in this paper. Since the alteration cost is user-specified, in certain problems, if the quadratic cost is not appropriate, one can replace it with any other convex functions under the increasing marginal cost (IMC) property [32]. In those cases, the alteration selection method is a convex optimization as well, similar to Eq. (6). It is worth noting that our work mainly focuses on cases where samples are rare, thus only user-specified hyperparameters $\{w_j, z_j^0\}_{j \in [|\mathbf{V}|]}$ can be relied upon. If we can obtain a sufficient number of $\langle$alteration, cost$\rangle$ pairs, hyperparameters $\{w_j, z_j^0\}_{j \in [|\mathbf{V}|]}$ can be accurately estimated using quadratic fitting techniques [16].

Additionally, for the probabilistic constraint in Eq. (2), if the noise is not assumed to be normal anymore, one can analyze the CDF of the noise distribution to find a new probability region with a similar approach. This can further assist in constructing a substitute constraint. Moreover, there are instances where the feasible domain of the constraint in Eq. (6) may not exist due to its stringent nature, *e.g.*, one aims to completely avoid undesired futures with 100% probability, or the desired region $\mathcal{S}$ is not practically achievable. In such cases, the decision-maker should consider verifying specified region $\mathcal{S}$ or reducing hyperparameter $\tau$ to some extent.

Last but not least, though we mainly focus on the scenario where the graph structure is fixed and known, *i.e.*, $G_t = G$, however, our approach can be straightforwardly adapted to the case where $G_t = G$ but unknown as assumed in Qin et al. [40] as detailed in Appendix A.3. We also provide a simple comparable experiment on Bermuda data in the setting of Qin et al. [40], as illustrated in Appendix D.1.

## A.2 Comparation with causality

The SRM and $Rh(\cdot)$ operations are similar to their counterparts in causal inference but have different applicating scenarios. The differences between rehearsal learning and causal learning are listed as follows:

1. Most of the causal learning problems focus on structure or effect identification, while the decision-making process is not directly involved in the modeling. In contrast, rehearsal learning mainly focuses on a class of decision-making problems that specializes in the goal of avoiding undesired future using SRM-based modeling.

2. Causal learning utilizes the SCM. In contrast, rehearsal learning uses the SRM (a new probabilistic graphical model [40]) to model the influence relations between the variables toward addressing the AUF problem. The modeling granularity of SRM is more flexible, as the influence relationship can be evolved, and mutually influenced. Specifically,

    - Possible dynamic influence relationships. For instance, the influence relationship between pricing and sales of coats can vary cyclically and trend-wise over time. Thus, a given price may result in fewer sales during summer compared to winter. In addition, the coat's style may gradually become outdated, leading to reduced sales at the same price a year later compared to the present.

    - Possible mutually influenced relationships. For instance, the ideal gas law states that the pressure $p$, volume $V$, amount of substance $n$, and absolute temperature $T$ obey the equation $pV = nRT$ ($R$ is the ideal gas constant). When a fixed volume $V$ of an ideal gas is considered, with $n$ and $R$ held constant, the pressure $p$ and volume $V$ can be represented as a pair of mutually influenced variables within an SRM modeling.

## A.3 Possible types of graph structures

There are four possible cases of the graph structure, specifically:

1. $G_t$ fixed, known. It's the setting in our paper.

2. $G_t$ fixed, unknown. It's the setting in Qin et al. [40]. Our method can be straightforwardly adapted to this case. Technically, since the distribution $\mathbb{P}(G)$ is discrete, we can initialize and update the probability mass function (PMF) $\mathbb{P}(G)$ in the same manner as Qin et al. [40]. Thus, when we want to select decision alterations, the maintained PMF $\mathbb{P}(G)$ can be utilized to marginalize $G$ from $\mathbb{P}(\mathbf{Y} \in \mathcal{S} \mid G, \cdots)$. thereby the expectation of matrices $\mathbf{A}, \mathbf{B}, \mathbf{C}$ in Eq. (6) can be obtained and used.

3. $G_t$ not fixed, known. Our method can be directly used because once the graph structure is known, $G$ is not a stochastic component. Thus, matrices $\mathbf{A}, \mathbf{B}, \mathbf{C}$ in Eq. (6) can be obtained in the same manner. It

is worth noting that the changing graph structure will lead to the birth of new parameters at each round. Therefore, though the method can be directly used in such case, establishing theoretical guarantees is difficult.

4 $G_t$ not fixed, unknown. In this case, neither $\theta$ nor $P_G$ can be accurately estimated as there is only one sample per round. Thus, dealing with such a case may need to additional assumptions. For example, consider there are a sufficient number of samples per round. In this case, our method can model $\boldsymbol{\theta}$ and $P_G$ accurately, thereby suggesting good decisions.

# B  Definitions

## B.1  Details about Structural Rehearsal Models

In this section, we present full definitions and discussions for the Structural Rehearsal Model (SRM), which is a probabilistic graphical model proposed by [40] to characterize the influence relations among variables.

### ● Definition of the Rehearsal Graph

**Definition B.1** (Mixed graph, [40]). Let $G = (\mathbf{V}, \mathbf{E})$ be a graph, where $\mathbf{V}$ denotes the vertices and $\mathbf{E}$ the edges. $G$ is a mixed graph if for any distinct vertices $u, v \in \mathbf{V}$, there is at most one edge connecting them, and the edge is either directional ( $u \to v$ or $u \leftarrow v$ ) or bi-directional ($u \leftrightarrow v$).

**Definition B.2** (Bi-directional clique, [40]). A bi-directional clique $C = (\mathbf{V}^c, \mathbf{E}^c)$ of a mixed graph $G = (\mathbf{V}, \mathbf{E})$ is a complete subgraph induced by $\mathbf{V}^c \subseteq \mathbf{V}$ such that any edge $e \in \mathbf{E}^c$ is bi-directional. $C$ is maximal if adding any other vertex does not induce a bi-directional clique.

**Definition B.3** (Rehearsal graph, [40]). Let $G = (\mathbf{V}, \mathbf{E})$ be a mixed graph. Let $\{C_i\}_{i=1}^l$ denote all maximal bi-directional cliques of $G$, where $C_i = (\mathbf{V}_i^c, \mathbf{E}_i^c)$. $G$ is a rehearsal graph if and only if:

1. $\mathbf{V}_i^c \cap \mathbf{V}_j^c = \emptyset$ for any $i \neq j$.

2. $\forall i \in [l]$, if there is any edge pointing from some $u \in \mathbf{V} \backslash \mathbf{V}_i^c$ to some $v \in \mathbf{V}_i^c$, then $\forall v \in \mathbf{V}_i^c$, $u \to v$.

3. There exists a topological ordering for $\{C_i\}_{i=1}^l$ following the directions of directional edges between $C_i$s.

It can be found that the topological ordering for bi-directional cliques $\{C_i\}_{i=1}^l$ in the rehearsal graph reflects the time order of the generation process of variables.

### ● Discussion of the Structural Equations

In this paper, we present the definition of dynamic structural equations in Eq. (1), which provides a quantitative computational formulation for the influence relations in non-stationary environments.

In fact, the structural equations are defined among the bi-directional cliques $\{C_i\}_{i=1}^l$ as detailed in Qin et al. [40]. Specifically, the dynamic structural equations can be denoted as $\boldsymbol{\theta}_t$, which consists of the set of parameter matrices $\{\boldsymbol{\beta}_{i,t}\}_{t=1}^T$ and the covariance matrix $\boldsymbol{\Sigma}_i$ of each clique $C_i$ in the rehearsal graph $G$. It is noteworthy that the quantitative influence relationship of a directional edge $A \to B$ for $A \in C_a, B \in C_b$ at time $t$ is modeled in the parameter matrix $\boldsymbol{\beta}_{b,t}$ since the topological ordering for bi-directional cliques reflects the time order of the generation process; while the quantitative influence relationship of a bi-directional edge $D_1 \leftrightarrow D_2$ for $D_1, D_2 \in C_d$ at time $t$ is modeled in the covariance matrix $\boldsymbol{\Sigma}_d$ since they are in the same bi-directional clique which is viewed as happening in the same time.

We present a simplified version of the definition of dynamic structural equations in Eq. (1) for a convenient presentation, which defines the dynamic structural equations on the variable level rather than the clique level. Note that for each variable $V_j \in C_i$, the associated parameter vectors $\{\beta_{j,t}\}_{t=1}^T$ can be directly found in the corresponding parameter matrices $\{\boldsymbol{\beta}_{i,t}\}_{t=1}^T$; while the variance $\sigma_j^2$ of each variable $V_j \in C_i$ can be derived from the corresponding covariance matrix $\boldsymbol{\Sigma}_i$ by marginalizing the desired dimension out, since the marginalization operator for multi-normal distribution is available.

## B.2  Increasing Marginal Cost with an Example

In this section, we introduce the increasing marginal cost (IMC) property. Increasing marginal cost (IMC), or rising marginal cost, is an important property in microeconomics [23, 32]. We first present a famous example in the field of microeconomics, called Thirsty Thelma's Lemonade Stand.

Lemonade Stand is a business simulation game devised in 1973 by Bob Jamison under the Minnesota Educational Computing Consortium. It offers players the experience of managing a lemonade stand across multiple rounds. At the commencement of each round, players make decisions regarding their stock, pricing, and advertising based on their current financial standing. The outcomes in each round are determined by the player's choices; and are further influenced by random events like thunderstorms and street closures. After each round, a summary of the current status of the player is provided, and the game concludes after 12 rounds. In this game, an essential quantity is the total cost of lemonade per glass, which will increase in a quadratic way, as it has a linearly increasing derivative (named the marginal cost) as shown in Fig. 5.

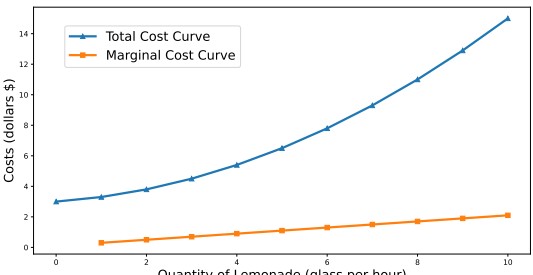

Figure 5: Thirsty Thelma's Total-Cost and Marginal-Cost Curves

This phenomenon is commonly observed in real-world scenarios, and the upward slope of the total cost curve reflects the property of the law of diminishing marginal product [32]. When Thirsty Thelma produces a small quantity of lemonade, her workforce is limited, and much of her equipment remains idle. The ease of utilizing these unused resources results in a substantial marginal product for an additional worker and a corresponding small marginal cost for producing an extra glass of lemonade. In contrast, as she increases lemonade production, her shop becomes crowded with workers, and most of her equipment is fully utilized. While additional workers can contribute to increased production, they must operate in crowded conditions and may face delays in equipment usage. Consequently, when the quantity of lemonade produced is already high, the marginal product of an extra worker diminishes, leading to a significant increase in the marginal cost of producing an extra glass of lemonade. It is a famous and widely used example in the field of microeconomics, and it has many similarities with the AUF scenario. In our proposed cost-minimal AUF problem, when the decision-maker decides to alter some variables in $\mathbf{Z}_t$, the cost associated with different alterations vary. For the same variable, the cost of altering it from a datum point should have a similar shape to the total cost curve in Fig. 5, varying with an increasing derivative. That is how we define the cost function in this paper. The cost function could be replaced by any other convex function in different situations.

## C   Proofs

In this section, we provide proof for claims in the main text.

### C.1   Proof for Proposition 3.2

**Lemma C.1.** *If $\mathbf{\Sigma}$ is a symmetric positive definite matrix, the following two sets are equivalent:*

$$\left\{\mathbf{x} : \mathbf{x}^\top \mathbf{\Sigma}^{-1} \mathbf{x} \leq 1\right\} \Leftrightarrow \left\{\mathbf{x} : \mathbf{x} = \mathbf{\Sigma}^{\frac{1}{2}} \mathbf{u} \mid \|\mathbf{u}\|_2 \leq 1\right\},$$

*where $\mathbf{x}$ and $\mathbf{u}$ are vectors with a center at the origin of the coordinate system $O$.*

*Proof.* Assume $\mathcal{P}_1 = \left\{\mathbf{x} : \mathbf{x}^\top \mathbf{\Sigma}^{-1} \mathbf{x} \leq 1\right\}$ and $\mathcal{P}_2 = \left\{\mathbf{x} : \mathbf{x} = \mathbf{\Sigma}^{\frac{1}{2}} \mathbf{u} \mid \|\mathbf{u}\|_2 \leq 1\right\}$. Then what we need to prove is equivalent to:

  (1)  Points contained in $\mathcal{P}_1$ have to be contained in $\mathcal{P}_2$.

  (2)  Points contained in $\mathcal{P}_2$ have to be contained in $\mathcal{P}_1$.

For (1), it holds that $\mathbf{x}^\top \mathbf{\Sigma}^{-1} \mathbf{x} \leq 1$, which is equivalent to $(\mathbf{\Sigma}^{-\frac{1}{2}} \mathbf{x})^\top (\mathbf{\Sigma}^{-\frac{1}{2}} \mathbf{x}) \leq 1$. Thus, let $\mathbf{u} = \mathbf{\Sigma}^{-\frac{1}{2}} \mathbf{x}$, it can be derived that $\mathbf{x} = \mathbf{\Sigma}^{\frac{1}{2}} \mathbf{u}$ where $\|\mathbf{u}\|_2 \leq 1$, and (1) is proved.

For (2), we replace $\mathbf{x}^\top \mathbf{\Sigma}^{-1} \mathbf{x}$ with $\mathbf{x} = \mathbf{\Sigma}^{\frac{1}{2}} \mathbf{u}$, then it can be derived that:

$$\mathbf{x}^\top \mathbf{\Sigma}^{-1} \mathbf{x} = \mathbf{u}^\top \mathbf{\Sigma}^{\frac{1}{2}} \mathbf{\Sigma}^{-1} \mathbf{\Sigma}^{\frac{1}{2}} \mathbf{u} = \mathbf{u}^\top \mathbf{u} \leq 1,$$

which shows that points contained in $\mathcal{P}_2$ are also contained in $\mathcal{P}_1$, so (2) is proved. In conclusion, the lemma is proved. □

**Proposition 3.2.** *The following set $\mathcal{P}$ is a probability region that satisfies $\mathbb{P}(\mathbf{Y}_t \in \mathcal{P} \mid \boldsymbol{\theta}_t, \mathbf{x}_t, Rh(\xi_t)) = \tau$:*

$$\mathcal{P} = \left\{ \boldsymbol{\mu}_{\mathbf{y}_t} + \left( \chi^{-1}(\tau) \mathbf{C}\boldsymbol{\Sigma}\mathbf{C}^\top \right)^{\frac{1}{2}} \mathbf{u} \;\middle|\; \|\mathbf{u}\|_2 \leq 1 \right\},$$

*where $\boldsymbol{\mu}_{\mathbf{y}_t} = \mathbf{A}\mathbf{x}_t + \mathbf{B}\mathbf{z}_t^\xi$, $\mathbf{u}$ is an arbitrary point in the unit sphere in $\mathbb{R}^{|\mathbf{Y}_t|}$, and $\chi^{-1}(\cdot)$ denotes the quantile function of the $\chi^2$ distribution with degrees of freedom $\lambda = |\mathbf{Y}_t|$.*

*Proof.* Recall from Lemma 3.1 that:

$$\boldsymbol{\varepsilon}_t \sim \mathcal{N}(\mathbf{0}, \boldsymbol{\Sigma}), \quad \mathbf{Y}_t = \mathbf{A}\mathbf{x}_t + \mathbf{B}\mathbf{z}_t^\xi + \mathbf{C}\boldsymbol{\varepsilon}_t.$$

From the property of multi-normal distributions, it holds that:

$$\mathbf{Y}_t \sim \mathcal{N}\left( \mathbf{A}\mathbf{x}_t + \mathbf{B}\mathbf{z}_t^\xi, \mathbf{C}\boldsymbol{\Sigma}\mathbf{C}^\top \right).$$

Let $\boldsymbol{\mu}_{\mathbf{y}_t} = \mathbf{A}\mathbf{x}_t + \mathbf{B}\mathbf{z}_t^\xi$, then we normalize the distribution above, it can be derived that:

$$\left( \mathbf{C}\boldsymbol{\Sigma}\mathbf{C}^\top \right)^{-\frac{1}{2}} (\mathbf{Y}_t - \boldsymbol{\mu}_{\mathbf{y}_t}) \sim \mathcal{N}(\mathbf{0}, \mathbf{I}).$$

So it can be derived that:

$$(\mathbf{Y}_t - \boldsymbol{\mu}_{\mathbf{y}_t})^\top \left( \mathbf{C}\boldsymbol{\Sigma}\mathbf{C}^\top \right)^{-1} (\mathbf{Y}_t - \boldsymbol{\mu}_{\mathbf{y}_t}) \sim \chi_\lambda^2,$$

where the $\chi^2$ distribution has the degrees of freedom $\lambda = |\mathbf{Y}_t|$.

Let $\chi^{-1}(\cdot)$ denote the quantile function of the distribution $\chi_\lambda^2$ above, then $\mathcal{P}$ can be given as:

$$\mathcal{P} = \left\{ \boldsymbol{\gamma} : (\boldsymbol{\gamma} - \boldsymbol{\mu}_{\mathbf{y}_t})^\top \left( \mathbf{C}\boldsymbol{\Sigma}\mathbf{C}^\top \right)^{-1} (\boldsymbol{\gamma} - \boldsymbol{\mu}_{\mathbf{y}_t}) \leq \chi^{-1}(\tau) \right\},$$

which is equivalent to:

$$\mathcal{P} = \left\{ \boldsymbol{\gamma} : (\boldsymbol{\gamma} - \boldsymbol{\mu}_{\mathbf{y}_t})^\top \left( \chi^{-1}(\tau)\mathbf{C}\boldsymbol{\Sigma}\mathbf{C}^\top \right)^{-1} (\boldsymbol{\gamma} - \boldsymbol{\mu}_{\mathbf{y}_t}) \leq 1 \right\},$$

By lemma C.1, it is equivalent to:

$$\mathcal{P} = \left\{ \boldsymbol{\gamma} : \boldsymbol{\gamma} = \boldsymbol{\mu}_{\mathbf{y}_t} + \left( \chi^{-1}(\tau)\mathbf{C}\boldsymbol{\Sigma}\mathbf{C}^\top \right)^{\frac{1}{2}} \mathbf{u} \;\middle|\; \|\mathbf{u}\|_2 \leq 1 \right\},$$

where $\boldsymbol{\mu}_{\mathbf{y}_t} = \mathbf{A}\mathbf{x}_t + \mathbf{B}\mathbf{z}_t^\xi$, and $\mathbf{u}$ is an arbitrary point in the unit sphere in $\mathbb{R}^{|\mathbf{Y}_t|}$. □

## C.2 Proof for Theorem 3.3

**Lemma C.2** (Theorem 3 of Cutler et al. [11]). *Consider a sequence of stochastic optimization problems $\min_\beta f_t(\beta)$ indexed by time $t \in \mathbb{N}$. Let $\hat{g}_t$ denote the estimation for the gradient of $f_t$, and $\beta_t^\star$ denote the minimizer of the L-smooth and $\mu$-strongly convex function $f_t$. Suppose it holds that $\max_t \hat{g}_t \leq \sigma^2 < \infty$ and $\max_t \mathbb{E}\|\beta_{t+1}^\star - \beta_t^\star\| \leq \Delta < \infty$. Then the produced $\{\beta_t\}_{t=0}^T$ with iterates $\beta_{t+1} = \Pi_\mathcal{B}(\beta_t - \eta\hat{g}_t)$ $(\mathcal{B} = \mathbb{R}^{|\beta|})$ and constant learning rate $\eta \leq 1/2L$ satisfies:*

$$\mathbb{E}\|\beta_t - \beta_t^\star\|^2 \lesssim \underbrace{(1 - \mu\eta)^t \|\beta_0 - \beta_0^\star\|^2}_{optimization} + \underbrace{\frac{\eta\sigma^2}{\mu}}_{noise} + \underbrace{\left( \frac{\Delta}{\mu\eta} \right)^2}_{drift}.$$

**Lemma C.3** (Convergence of LSE). *Let $\beta_{j,t}$ denote the true parameter value of $\beta_j$ in time t, and $\hat{\beta}_{j,t}^{lse}$ denote the estimation according to Eq. (4). For $\forall \psi > 0$, $\exists n > 0$ that holds:*

$$\mathbb{E}\left\| \hat{\beta}_{j,t}^{lse} - \beta_{j,t} \right\|_2^2 \leq \psi$$

*Proof.* Let $\mathbf{P}_{j,t} = \begin{bmatrix} \mathrm{PA}_{j,t}^1 & \cdots & \mathrm{PA}_{j,t}^n \end{bmatrix}^\top$ and $\mathbf{v}_{j,t} = \begin{bmatrix} V_{j,t}^1, \ldots, V_{j,t}^n \end{bmatrix}^\top$. Then the optimization in Eq. (4) is equivalent to:

$$\arg\min_{\beta_{j,t}} \|\mathbf{v}_{j,t} - \mathbf{P}_{j,t}\beta_{j,t}\|_2^2.$$

Since $\mathrm{PA}_j$ represents the true parents of the generation process for variable $V_j$, $\mathbf{P}_{j,t}$ is guaranteed to have full column rank. For $\forall n \geq |\mathrm{PA}_j|$, the solution to the optimization above is:

$$\hat{\beta}_{j,t}^{lse} = \left(\mathbf{P}_{j,t}^\top \mathbf{P}_{j,t}\right)^{-1} \mathbf{P}_{j,t}^\top \mathbf{v}_{j,t}.$$

Because $\mathbf{v}_{j,t} = \mathbf{P}_{j,t}\beta_{j,t} + \varepsilon_j$ where $\varepsilon_j \sim \mathcal{N}\left(\mathbf{0}, \sigma_j^2 \mathbf{I}_n\right)$ according to Eq. (3), we have:

$$\hat{\beta}_{j,t}^{lse} = \beta_{j,t} + \left(\mathbf{P}_{j,t}^\top \mathbf{P}_{j,t}\right)^{-1} \mathbf{P}_{j,t}^\top \varepsilon_j.$$

Due to the linear combination property of the Gaussian distribution, it holds that:

$$\hat{\beta}_{j,t}^{lse} - \beta_{j,t} \sim \mathcal{N}\left(\mathbf{0}, \sigma_j^2 \left(\mathbf{P}_{j,t}^\top \mathbf{P}_{j,t}\right)^{-1}\right)$$

Thus, it can be derived that:

$$
\begin{aligned}
\mathbb{E}\left\|\hat{\beta}_{j,t}^{lse} - \beta_{j,t}\right\|_2^2 &= \mathrm{tr}\left(\sigma_j^2 \left(\mathbf{P}_{j,t}^\top \mathbf{P}_{j,t}\right)^{-1}\right) \\
&= \sigma_j^2 \sum_{i=1}^{|\mathrm{PA}_j|} \lambda_i \left(\lambda_i\text{s are eigenvalues of } \left(\mathbf{P}_{j,t}^\top \mathbf{P}_{j,t}\right)^{-1}\right) \\
&\leq \lambda_{max} \cdot \sigma_j^2 |\mathrm{PA}_j| = \left\|\left(\mathbf{P}_{j,t}^\top \mathbf{P}_{j,t}\right)^{-1}\right\|_2 \cdot \sigma_j^2 |\mathrm{PA}_j|
\end{aligned}
$$

Because $\left\|\left(\mathbf{P}_{j,t}^\top \mathbf{P}_{j,t}\right)^{-1}\right\|_2 \cdot \left\|\mathbf{P}_{j,t}^\top \mathbf{P}_{j,t}\right\|_2 \leq \left\|\left(\mathbf{P}_{j,t}^\top \mathbf{P}_{j,t}\right)^{-1} \mathbf{P}_{j,t}^\top \mathbf{P}_{j,t}\right\|_2 = 1$, we have:

$$
\begin{aligned}
\mathbb{E}\left\|\hat{\beta}_{j,t}^{lse} - \beta_{j,t}\right\|_2^2 &\leq \frac{\sigma_j^2 |\mathrm{PA}_j|}{\left\|\mathbf{P}_{j,t}^\top \mathbf{P}_{j,t}\right\|_2} \\
&\leq \frac{\sigma_j^2 |\mathrm{PA}_j|^2}{\sum_{i=1}^{|\mathrm{PA}_j|} \gamma_i} \left(\gamma_i\text{s are eigenvalues of } \left\|\mathbf{P}_{j,t}^\top \mathbf{P}_{j,t}\right\|_2\right) \\
&= \frac{\sigma_j^2 |\mathrm{PA}_j|^2}{\mathrm{tr}\left(\mathbf{P}_{j,t}^\top \mathbf{P}_{j,t}\right)} = \frac{\sigma_j^2 |\mathrm{PA}_j|^2}{\sum_{i=1}^{|\mathrm{PA}_j|}\sum_{k=1}^{n} p_{ik}^2} \leq \frac{\sigma_j^2 |\mathrm{PA}_j|^2}{n \min_k \sum_{i=1}^{|\mathrm{PA}_j|} p_{ik}^2}
\end{aligned}
$$

So for $\forall \psi > 0$, $\exists n = \sigma_j^2 |\mathrm{PA}_j|^2 \left(\psi \min_k \sum_{i=1}^{|\mathrm{PA}_j|} p_{ik}^2\right)^{-1}$ that holds $\mathbb{E}\left\|\hat{\beta}_{j,t}^{lse} - \beta_{j,t}\right\|_2^2 \leq \psi$. □

**Lemma C.4** (Positive-definite Property). *Consider a set of $m$-dimensional vectors $\{\mathbf{v}_i\}_{i=1}^n$, $n \geq m$. If it contains at least one set of basis vectors in space $\mathbb{R}^m$, then the following matrix is positive-definite:*

$$\mathbf{M} = \sum_{i=1}^n \mathbf{v}_i \mathbf{v}_i^\top$$

*Proof.* Let $\{\mathbf{u}_k\}_{k=1}^m \subseteq \{\mathbf{v}_i\}_{i=1}^n$ denote a set of basis vectors, and let $\mathbf{x} \in \mathbb{R}^m$ denote an arbitrary $m$-dimensional points in the same space as $\mathbf{v}_i$s. Then it can be derived that:

$$
\begin{aligned}
\mathbf{x}^\top \mathbf{M} \mathbf{x} &= \sum_{i=1}^n \left(\mathbf{x}^\top \mathbf{v}_i\right)^2 \\
&\geq \sum_{k=1}^m \left(\mathbf{x}^\top \mathbf{u}_k\right)^2 > 0,
\end{aligned}
$$

because there at least exists one basis vector $\mathbf{u}_h$ that holds $\mathbf{x}^\top \mathbf{u}_h \neq 0$. So $\mathbf{M}$ is positive-definite (thus $\mathbf{M}$ is full-rank). □

**Theorem 3.3.** *Denote the true parameter of $\beta_j$ in time $t$ as $\beta_{j,t}$ ($j \in [|\mathbf{V}|]$) and choose $\eta_j \in (0, 1/2L_j]$ as the step size used in Algorithm 1, it can be bounded in G that:*

$$\mathbb{E}\left\|\hat{\beta}_{j,t} - \beta_{j,t}\right\|^2 \lesssim (1 - \mu_j \eta_j)^{\frac{t}{m}} \left\|\hat{\beta}_{j,0} - \beta_{j,0}\right\|^2 + \delta_j \quad \text{with} \quad \delta_j = \left(\frac{m\Delta_j}{\mu_j \eta_j}\right)^2 + \frac{\eta_j \sigma^2}{\mu_j},$$

*where $\mu_j$ and $L_j$ are the minimal and maximal eigenvalues of $\{\ell_{j,t}(\cdot)\}_{t=1}^T$'s Hessian matrices respectively, $\sigma^2$ upper-bounds the variance of $\hat{g}_{j,t}$ used in Algorithm 1, $\Delta_j \geq \max_t \|\beta_{j,t+1} - \beta_{j,t}\|$ upper-bounds the varying speed of the environment, and $m$ is the longest continuously altered rounds of $V_j$, for most of the $V_j$s, $m = 1$.*

*Proof.* For Eq. (4), the Hessain matrix of $\ell_{j,t}(\cdot)$ can be derived that:

$$\mathbf{H}_{j,t} = \nabla^2 \ell_{j,t} = \frac{1}{n}\sum_{k=1}^{n} \mathrm{PA}_{j,t}^k\,\mathrm{PA}_{j,t}^{k\ \top}$$

Since $n \gg |\mathrm{PA}_j|$ and $\mathrm{PA}_{j,t}^k$s are $n$ discrete samples from a continuous distribution, so $\{\mathrm{PA}_{j,t}^k\}_{k=1}^{n}$ contains a set of basis vectors in space $\mathbb{R}^{|\mathrm{PA}_j|}$ with probability 1. By Lemma C.4, $\{\mathbf{H}_{j,t}\}_{t=1}^{T}$ are positive-definite.

Let $\mu_j$ denote $\min_t \lambda(\mathbf{H}_{j,t})$ and $L_j$ denote $\max_t \lambda(\mathbf{H}_{j,t})$ respectively, then $\{\ell_{j,t}(\cdot)\}_{t=1}^{T}$ are all $L_j$-smooth and $\mu_j$-strongly convex. By Lemma C.2 and consider alterations on $V_j$, estimations $\left\{\hat{\beta}_{j,t}\right\}_{t=1}^{T}$ obtained from Algorithm 1 hold that:

$$\mathbb{E}\left\|\hat{\beta}_{j,t} - \beta_{j,t}^\star\right\|^2 \lesssim (1 - \mu_j\eta_j)^{t/m}\left\|\hat{\beta}_{j,0} - \beta_{j,0}^\star\right\|^2 + \left(\frac{m\hat{\Delta}_j}{\mu_j\eta_j}\right)^2 + \frac{\eta_j\sigma^2}{\mu_j};$$

where $\sigma^2$ upper-bounds the variance of $\hat{g}_{j,t}$ used in Algorithm 1, $\hat{\Delta}_j \geq \max_t \|\beta_{j,t+1}^\star - \beta_{j,t}^\star\|$ upper-bounds the varying speed of the minimizer of $\ell_{j,t}$, and $m$ is the longest continuously altered rounds of $V_j$, for most of the $V_j$s, $m = 1$. This holds because $m\hat{\Delta}_j \geq m\max_t \|\beta_{j,t+1}^\star - \beta_{j,t}^\star\| \geq \max_t \sum_{k=1}^{m}\|\beta_{j,t+k}^\star - \beta_{j,t+k-1}^\star\| \geq \max_t \|\beta_{j,t+m}^\star - \beta_{j,t}^\star\|$. Note that if $V_j$ is continuously altered $\leq m$ rounds, it can be viewed as its corresponding parameters are updated per $m$ rounds.

Besides, since $\hat{\beta}_{j,t}^{lse} \triangleq \beta_{j,t+1}^\star$ in such case, let $\Delta_j \geq \max_t \|\beta_{j,t+1}^\star - \beta_{j,t}^\star\|$, by Lemma C.3, it holds that $\hat{\Delta}_j \lesssim \Delta_j$, $\left\|\hat{\beta}_{j,0} - \beta_{j,0}^\star\right\|^2 \lesssim \left\|\hat{\beta}_{j,0} - \beta_{j,0}\right\|^2$, $\mathbb{E}\left\|\beta_{j,t}^\star - \beta_{j,t}\right\|^2 \leq \psi$; and we have:

$$\mathbb{E}\left\|\hat{\beta}_{j,t} - \beta_{j,t}\right\|^2 \lesssim \mathbb{E}\left\|\hat{\beta}_{j,t} - \beta_{j,t}^\star\right\|^2 + \mathbb{E}\left\|\beta_{j,t}^\star - \beta_{j,t}\right\|^2$$

$$\lesssim (1 - \mu_j\eta_j)^{t/m}\left\|\hat{\beta}_{j,0} - \beta_{j,0}\right\|^2 + \left(\frac{m\Delta_j}{\mu_j\eta_j}\right)^2 + \frac{\eta_j\sigma^2}{\mu_j}$$

$\square$

## C.3   Proof of Proposition 3.4

**Lemma C.5** (Hoeffding, 1963)**.** *Let $X$ be a random variable with $a \leq X \leq b$. Then, for any $s \in \mathbb{R}$,*

$$\ln \mathrm{E}\left[e^{sX}\right] \leq s\mathrm{E}[X] + \frac{s^2(b-a)^2}{8}.$$

**Proposition 3.4.** *Assume $\{\ell_{j,t}(\cdot)\}_{t=1}^{T}$s are bounded for $\forall \beta_i \in \mathcal{B}$ and $t \in [T]$; then for any $\eta \in \mathcal{H}_j$, estimations $\hat{\beta}_{j,t}$s from Algorithm 2 satisfies that*

$$\sum_{t=1}^{T}\hat{\ell}_t\left(\hat{\beta}_{j,t}\right) - \sum_{t=1}^{T}\hat{\ell}_t\left(\beta_{j,t}^\eta\right) \leq \mathcal{O}\left(\sqrt{T\ln N_j}\right);$$

*by choosing $\alpha = \sqrt{\ln N_j/T}$ in Algorithm 2, where $N_j$ is the number of base-learners, $\beta_{j,t}^\eta$ is the estimation from any expert $\eta$ in expert set $\mathcal{H}_j$ in Algorithm 2.*

*Proof.* Following previous studies [8] (Theorem 2.2 and Exercise 2.5), we define:

$$L_t^\eta = \sum_{i=1}^{t}\hat{\ell}_{j,i}\left(\beta_{j,i}^\eta\right), \text{ and } W_t = \sum_{\eta\in\mathcal{H}_j}w_1^\eta e^{-\alpha L_t^\eta}.$$

From the updating rule in Algorithm 2, it is easy to verify that:

$$w_t^\eta = \frac{w_1^\eta e^{-\alpha L_{t-1}^\eta}}{\sum_{\mu\in\mathcal{H}}w_1^\mu e^{-\alpha L_{t-1}^\mu}}, \ t \geq 2. \tag{7}$$

First, it can be derived that:

$$\ln W_T = \ln\left(\sum_{\eta\in\mathcal{H}_j}w_1^\eta e^{-\alpha L_T^\eta}\right) \geq \ln\left(\max_{\eta\in\mathcal{H}_j}w_1^\eta e^{-\alpha L_T^\eta}\right) = -\alpha\min_{\eta\in\mathcal{H}_j}\left(L_T^\eta + \frac{1}{\alpha}\ln\frac{1}{w_1^\eta}\right). \tag{8}$$

Next, the related quantity $\ln(W_t/W_{t-1})$ can be bounded as follows when $t \geq 2$:

$$\ln\left(\frac{W_t}{W_{t-1}}\right) = \ln\left(\frac{\sum_{\eta \in \mathcal{H}} w_1^\eta e^{-\alpha L_t^\eta}}{\sum_{\mu \in \mathcal{H}_j} w_1^\mu e^{-\alpha L_{t-1}^\mu}}\right)$$

$$= \ln\left(\sum_{\eta \in \mathcal{H}_j}\left(\frac{w_1^\eta e^{-\alpha L_{t-1}^\eta}}{\sum_{\mu \in \mathcal{H}_j} w_1^\mu e^{-\alpha L_{t-1}^\mu}} e^{-\alpha \hat\ell_{j,t}\left(\beta_{j,t}^\eta\right)}\right)\right)$$

$$\overset{(7)}{=} \ln\left(\sum_{\eta \in \mathcal{H}_j} w_t^\eta e^{-\alpha \hat\ell_{j,t}\left(\beta_{j,t}^\eta\right)}\right).$$

When $t = 1$, it holds that $\ln W_1 = \ln\left(\sum_{\eta \in \mathcal{H}} w_1^\eta e^{-\alpha \hat\ell_{j,1}\left(\beta_{j,1}^\eta\right)}\right)$, thus it can be derived that:

$$\ln W_T = \ln W_1 + \sum_{t=2}^{T} \ln\left(\frac{W_t}{W_{t-1}}\right) = \sum_{t=1}^{T} \ln\left(\sum_{\eta \in \mathcal{H}_j} w_t^\eta e^{-\alpha \hat\ell_{j,t}\left(\beta_{j,t}^\eta\right)}\right)$$

$$\leq -\alpha \sum_{\eta \in \mathcal{H}_j} w_t^\eta \hat\ell_{j,t}\left(\beta_{j,t}^\eta\right) + \frac{\alpha^2 c^2}{8} \quad (c \text{ is a constant as } \hat\ell_{j,t}(\cdot) \text{ is bounded})$$

$$\leq -\alpha \hat\ell_{j,t}\left(\sum_{\eta \in \mathcal{H}_j} w_t^\eta \beta_{j,t}^\eta\right) + \frac{\alpha^2 c^2}{8} = -\alpha \hat\ell_{j,t}\left(\hat\beta_{j,t}\right) + \frac{\alpha^2 c^2}{8},$$

$$(9)$$

where the inequality in the second line is due to Lemma C.5, and the inequality in the second line is due to Jensen's inequality. By combining Eq. (8) and Eq. (9), it can be derived that:

$$\sum_{t=1}^{T} \hat\ell_{j,t}\left(\hat\beta_{j,t}\right) - \min_{\eta \in \mathcal{H}_j}\left(\sum_{t=1}^{T} \hat\ell_{j,t}\left(\beta_{j,t}^\eta\right) + \frac{1}{\alpha}\ln\frac{1}{w_1^\eta}\right) \leq \frac{\alpha T c^2}{8}$$

Since we choose $w_1^\eta = \frac{1}{N_j}$ in Algorithm 2, thus for $\eta \in \mathcal{H}_j$, by choosing $\alpha = \sqrt{\ln N_j / T}$, it holds that:

$$\sum_{t=1}^{T} \hat\ell_{j,t}\left(\hat\beta_{j,t}\right) - \sum_{t=1}^{T} \hat\ell_{j,t}\left(\beta_{j,t}^\eta\right) \lesssim \alpha T + \frac{1}{\alpha}\ln N_j = \mathcal{O}\left(\sqrt{T \ln N_j}\right)$$

$\square$

## C.4 Proof for Theorem 3.5

**Theorem 3.5.** *By using the suggested alterations from Eq. (6), it can be guaranteed that:*

$$\mathbb{P}\left(\mathbf{Y}_t \in \mathcal{S} \mid \hat{\boldsymbol{\theta}}_t, \mathbf{x}_t, Rh(\xi_t)\right) \geq \tau.$$

*Proof.* Recall that Eq. (6) suggests alterations as follows:

$$\min_{\mathbf{z}_t^{\xi_t}} \quad \left(\mathbf{z}_t^{\xi_t} - \mathbf{z}_t^0\right)^\top \mathbf{W}\left(\mathbf{z}_t^{\xi_t} - \mathbf{z}_t^0\right)$$

$$\text{s.t.} \quad \mathbf{MAx}_t + \mathbf{MBz}_t^{\xi_t} + \|\mathbf{MP}\|_{2,\text{row}} \leq \mathbf{d},$$

where $\mathbf{P} = \left(\chi^{-1}(\tau)\mathbf{C}\boldsymbol{\Sigma}\mathbf{C}^\top\right)^{\frac{1}{2}}$, and $\|\cdot\|_{2,\text{row}}$ means an operator that takes 2-norm for each row of the matrix thus outputs a row-dimensional vector. We omit the subscript 2 of the norm in the following discussions.

Let $\boldsymbol{\gamma}_i$ denote the $i$-th row-vector of matrix $\mathbf{MP}$ and $r$ denote the row-dimension of $\mathbf{MP}$, then it can be derived that:

$$
\begin{aligned}
\mathbf{d} &\geq \mathbf{MAx}_t + \mathbf{MBz}_t^{\xi_t} + \|\mathbf{MP}\|_{2,\mathrm{row}} \\
&= \mathbf{MAx}_t + \mathbf{MBz}_t^{\xi_t} + \left(1 \cdot \|\boldsymbol{\gamma}_1\|, \quad \cdots, \quad 1 \cdot \|\boldsymbol{\gamma}_r\|\right)^\top \\
&\geq \mathbf{MAx}_t + \mathbf{MBz}_t^{\xi_t} + \left(\sup_{\|\mathbf{u}\|\leq 1} \|\boldsymbol{\gamma}_1\|\,\|\mathbf{u}\|\cos\langle\boldsymbol{\gamma}_1, \mathbf{u}\rangle, \quad \cdots, \quad \sup_{\|\mathbf{u}\|\leq 1} \|\boldsymbol{\gamma}_r\|\,\|\mathbf{u}\|\cos\langle\boldsymbol{\gamma}_r, \mathbf{u}\rangle\right)^\top \\
&= \mathbf{MAx}_t + \mathbf{MBz}_t^{\xi_t} + \left(\sup_{\|\mathbf{u}\|\leq 1} \langle\boldsymbol{\gamma}_1, \mathbf{u}\rangle, \quad \cdots, \quad \sup_{\|\mathbf{u}\|\leq 1} \langle\boldsymbol{\gamma}_r, \mathbf{u}\rangle\right)^\top \\
&= \sup_{\|\mathbf{u}\|\leq 1} \mathbf{MAx}_t + \mathbf{MBz}_t^{\xi_t} + \mathbf{MPu} \\
&= \sup_{\|\mathbf{u}\|\leq 1} \mathbf{M}\left(\boldsymbol{\mu}_{\mathbf{y}_t} + \left(\chi^{-1}(\tau)\mathbf{C\Sigma C}^\top\right)^{\frac{1}{2}}\mathbf{u}\right)
\end{aligned}
$$

where $\boldsymbol{\mu}_{\mathbf{y}_t} = \mathbf{Ax}_t + \mathbf{Bz}_t^{\xi_t}$. Recall from Prop. 3.2 that $\mathcal{P} = \left\{\boldsymbol{\mu}_{\mathbf{y}_t} + \left(\chi^{-1}(\tau)\mathbf{C\Sigma C}^\top\right)^{\frac{1}{2}}\mathbf{u}\right\}$ is the probability region that satisfies $\mathbb{P}\left(\mathbf{Y}_t \in \mathcal{S} \mid \hat{\boldsymbol{\theta}}_t, \mathbf{x}_t, Rh(\xi_t)\right) = \tau$ (since $\mathbf{A}, \mathbf{B}, \mathbf{C}$ are generated from $\hat{\boldsymbol{\theta}}_t$ rather than $\boldsymbol{\theta}_t$ in this case). Thus, Prop. 3.2 has been proved because (a) it has been derived that for $\forall \mathbf{y}_t \in \mathcal{P}$ it holds that $\mathbf{d} \geq \mathbf{My}_t$; and (b) the desired region $\mathcal{S}$ defined in Eq. (3) is $\mathcal{S} = \left\{\mathbf{y} \in \mathbb{R}^{|\mathbf{Y}|} \mid \mathbf{My} \leq \mathbf{d}\right\}$. $\qquad\square$

# D  Experimental details

The experiments are done by using macOS Monterey, Apple M1 Pro. All algorithms are running under the same environment.

## D.1  Comparison experiment in the setting of Qin et al. [40]

We provide a comparison experiment in the setting of Qin et al. [40], where $G \sim \mathbb{P}(G)$ and $\mathbb{P}(G)$ are unknown. Results on Bermuda data provided in the following table show that our approach exhibits a comparable performance with the result under the scenario where $G_t = G$.

| | Success Frequency | Cost | Time (s) |
|---|---|---|---|
| $G_t = G$ | $0.711 \pm 0.018$ | $1.46 \pm 0.05$ | $1.71 \pm 0.34$ |
| $G_t \sim \mathbb{P}_G$ | $0.698 \pm 0.021$ | $1.43 \pm 0.07$ | $1.99 \pm 0.46$ |

## D.2  Market-Manage Data

In this section, we provide details about the Market-Manage data. The variables included in the generation process are:

- Feature$_{\mathrm{our}}$: The feature used to predict the raw cost of our market;

- Feature$_{\mathrm{cpt}}$: The feature used to predict the raw cost of the competitor market;

- $C_{\mathrm{our}}$: The raw cost of our market;

- $C_{\mathrm{cpt}}$: The raw cost of the competitor market;

- $P_{\mathrm{our}}$: The product price of our market;

- $P_{\mathrm{cpt}}$: The product price of the competitor market;

- NCT: Customer numbers of our market;

- TPF: Total profit of our market.

The rehearsal graph for the variables is illustrated in Fig. 6. The presumed actionable variables that can be altered by the manager are $C_{\mathrm{our}}$ and $P_{\mathrm{our}}$. The hyperparameters associated with the cost function are set as $Z_{C_{\mathrm{our}}}^0 = 0.75$, $w_{C_{\mathrm{our}}} = 2.0$; and $Z_{P_{\mathrm{our}}}^0 = 0.0$, $w_{P_{\mathrm{our}}} = 1.0$. The desired region $\mathcal{S}$ is shown in Fig. 7(a). We shade dynamic edges with red color.

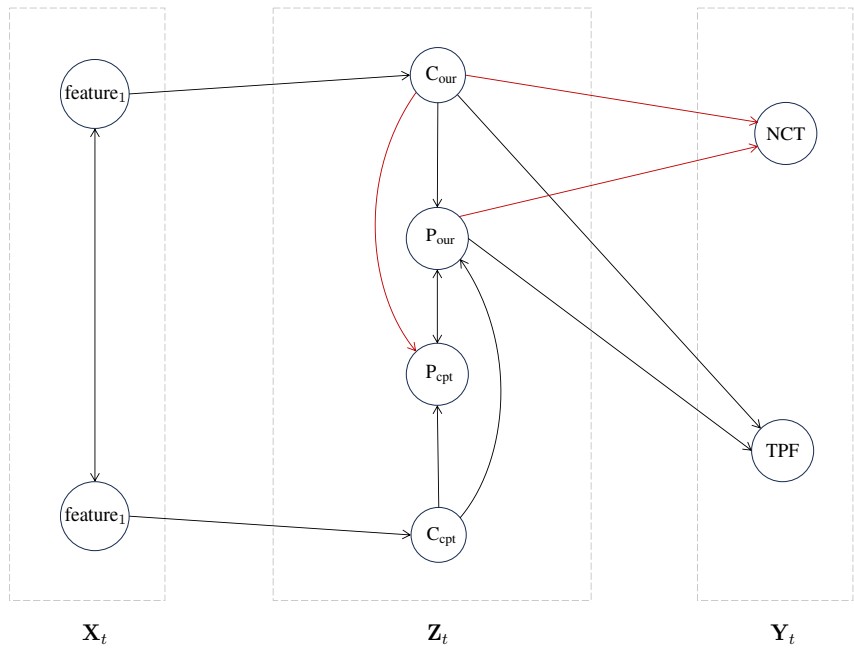

Figure 6: The rehearsal graph for market-manage data.

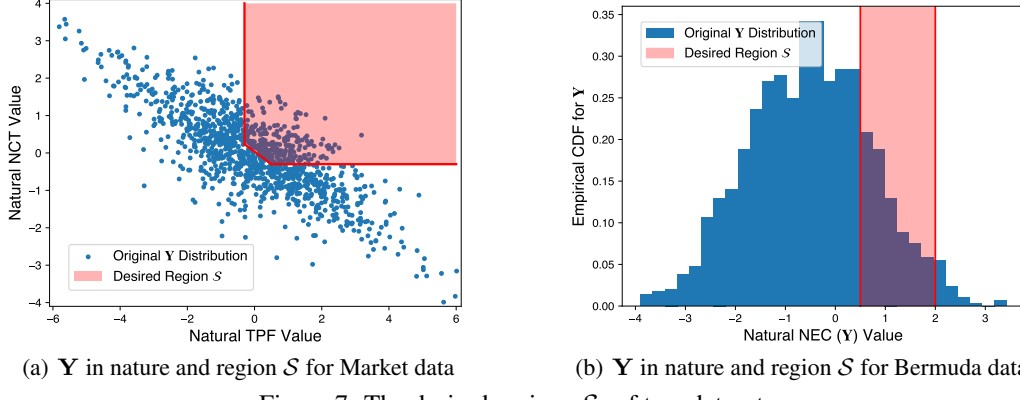

(a) $\mathbf{Y}$ in nature and region $\mathcal{S}$ for Market data  (b) $\mathbf{Y}$ in nature and region $\mathcal{S}$ for Bermuda data

Figure 7: The desired regions $\mathcal{S}$s of two datasets.

## D.3  Bermuda Data

In this section, we provide details about the Bermuda data. The Bermuda data is an environment dataset that involves some environmental variables in Bermuda [10]. The variables included in the generation process are:

- Light: Light levels at the bottom;
- Temp: Temperature at the bottom;
- Sal: Sea surface salinity;
- DIC: Dissolved inorganic carbon of seawater;
- TA: Total alkalinity of seawater;
- $\Omega_A$: Saturation with respect to aragonite in seawater;
- Chla: Chlorophyll-a at sea surface;
- Nut: PC1 of $NH_4$, $NiO_2 + NiO_3$, $SiO_4$;
- pHsw: pH of seawater;
- $CO_2$: $P_{CO_2}$ of seawater;
- NEC: Net ecosystem calcification.

The rehearsal graph for the variables is illustrated in Fig. 8. The presumed actionable variables that can be altered by the decision-maker are DIC, TA, $\Omega_A$, Chla, and Nut according to Aglietti et al. [1], Qin et al. [40]. The hyperparameters associated with the cost function are set as $Z^0_{\text{DIC}} = Z^0_{\text{TA}} = Z^0_{\Omega_A} = Z^0_{\text{Chla}} = Z^0_{\text{Nut}} = 0.0$; and $w_{\text{DIC}} = 10.0$, $w_{\text{TA}} = 8.0$, $w_{\Omega_A} = 3.0$, $w_{\text{Chla}} = 5.0$, $w_{\text{Nut}} = 10.0$. The desired region $\mathcal{S}$ is shown in Fig. 7(b). We shade dynamic edges with red color.

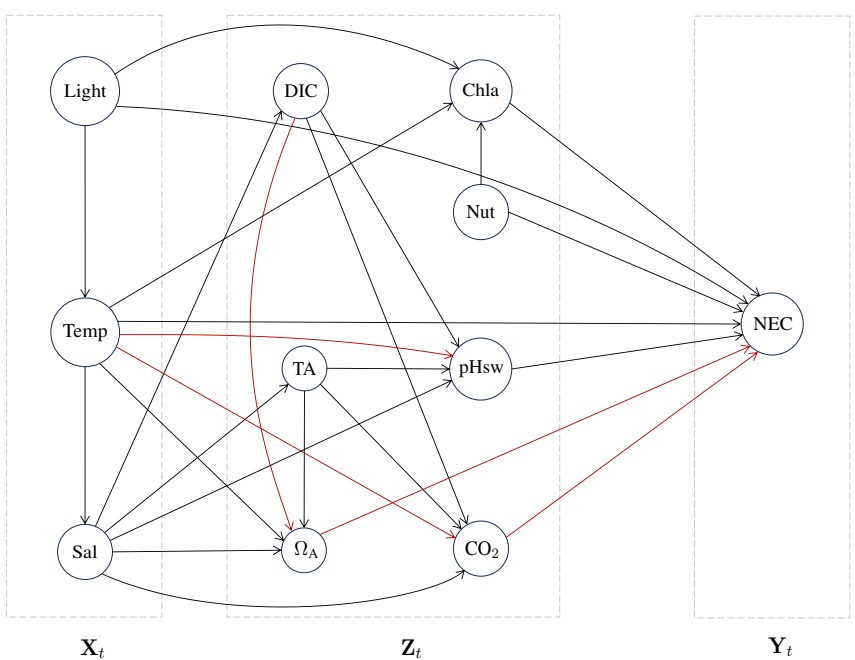

Figure 8: The rehearsal graph for Bermuda data.

