# OpenReview forum: "Avoiding Undesired Future with Minimal Cost in Non-Stationary Environments"
_NeurIPS.cc/2024/Conference — NeurIPS 2024 poster_

### Official Review · Reviewer_weAS · 2024-07-08

**Soundness:** 3
**Presentation:** 3
**Contribution:** 3
**Rating:** 5
**Confidence:** 3

**Summary:**

The paper studies the non-stationary setting in avoiding undesired future (AUF) problems, where environmental shifts can cause the failure of existing AUF methods. It introduces an optimization problem for AUF with minimal action cost in non-stationary environments, formulated as a convex quadratically constrained quadratic program (QCQP) in each interaction. The paper also proposes a rehearsal-based algorithm to solve this problem, providing theoretical guarantees and numerical validations.

**Strengths:**

The paper is well-written, introduces a practical and interesting setting for AUF problems, and presents an algorithm with theoretical guarantees and numerical validations to address the task.

**Weaknesses:**

(1) The provided algorithms lack a regret bound (or other theoretical guarantees) on the cost (i.e., the objective function), although it guarantees effective alterations (i.e., the constraint). Since the aim of this work is to avoid an undesired future with minimal cost, a regret bound analysis is, in my opinion, important.

(2) In Theorem 3.3, the estimation error depends on the minimum eigenvalues of  the empirical error functions' Hessian matrices, which in turn depends on the previously taken alterations. This raises a concern about the exploration-exploitation tradeoff when making alterations. An extreme case is making uninformative alterations (e.g., setting 0 for all nodes), leading to no update by Algorithm 1 and rendering the error bound in Theorem 3.3 meaningless (since $\mu_j$=0 in this case if I understand correctly). It is unclear how Algorithm 3 addresses this tradeoff and how $\mu_j$’s can be bounded  below.

**Questions:**

(1) How do algorithms handle the exploration-exploitation tradeoff (if explorations are needed)?

(2) Is it possible to establish a regret bound for the cost? If not, what are the challenges?

**Limitations:**

See weaknesses.

---

> ### Author Rebuttal · Authors · 2024-08-06
>
> Thanks for your detailed feedback, and we hope our responses will address your concerns.
>
> **W1&Q2.** Theoretical guarantees (regret bound) for the cost.
>
> **A1.** Thanks for your insightful question. In fact, theoretical guarantees of the cost can be inferred from existing results (Lemma 3.1 and Theorem 3.3), and we discuss as follows:
> - First, by using the interior-point method, Eq. (6) can achieve the optimal solution $z^*_{\hat{\theta}_t}$ that minimizes the objective cost function in each decision round $t$ since it is a convex QCQP [1, 2]. This is discussed in lines 265-271.
> - Because the constraint in Eq. (6) is constructed by $\hat{\theta_t}$ rather than $\theta_t$, let $C(\cdot)$ denote the cost function. The cost regret is $C(z^*_{\hat{\theta_t}}) - C(z^*_{\theta_t})$. Meanwhile, since $z^*(\cdot)$ can also be viewed as a function, let $f(\cdot)$ denote $C \circ z^*(\cdot)$. The regret for the cost is $f(\hat{\theta}_t) - f(\theta_t)$ in round $t$. Consider the following properties:
>     1. $||\hat{\theta}_t - \theta_t||^2$ enjoys a linear convergence rate as guaranteed in Theorem 3.3.
>     2. $f(\cdot)$ is a polynomial function, because $C(\cdot)$ is quadric (cost function) and $z^*(\cdot)$ is polynomial (as $z^*(\cdot)$ is polynomial w.r.t. $\mathbf{A}, \mathbf{B}, \mathbf{C}$ in the constraint of Eq. (6), and $\mathbf{A}, \mathbf{B}, \mathbf{C}$ are all polynomial w.r.t. $\theta_t$/$\hat{\theta}_t$, by proof process of Lemma. 3.1 [3]).
>     3. Both $||\hat{\theta}_t||$ and $||\theta_t||$ are bounded, by definition.
> - As implied by the properties above, $||f(\hat{\theta_t}) - f(\theta_t)||^2$ also enjoys a linear convergence rate. Consider the following simple example as an illustration:
>  $$
>     (x-y)^2\leq Le^{-t} \quad\mathop{\Longrightarrow}\quad (x^2-y^2)^2 =(x+y)^2(x-y)^2
>     \leq (|x|+|y|)^2(x-y)^2
>     \leq 4U^2L e^{-t} = \mathcal{O}(e^{-t}),
> $$
> where $U\in \mathbb{R}_*$ is the upper bound of $|x|$ and $|y|$.
>
> We will refine this discussion in the revised paper to make the guarantees more clear. Thanks!
>
> ---
>
> **W2.**  The minimal eigenvalue of the Hessian matrix in Theorem 3.3.
>
> **A2.** Thanks for your feedback. The Hessian matrix in Theorem 3.3 refers to $\nabla^2 \ell(\cdot)$ instead of $\nabla^2 \hat{\ell}(\cdot)$, where functions $\ell(\cdot)$ and $\hat{\ell}(\cdot)$ are defined in Eq. (4) and Eq. (5), respectively. Meanwhile, the $\hat{g}_{j,t}$ in Algorithm 1 refers to the gradient of $\hat{\ell}(\cdot)$, i.e., $\nabla \hat{\ell}(\cdot)$, instead of $\nabla \ell(\cdot)$.
>
> As a surrogate loss function of $\ell(\cdot)$, $\hat{\ell}(\cdot)$ only uses the collected sample in each round to approximate $\ell(\cdot)$. Hence, it is possible that the minimal eigenvalue of $\nabla^2 \hat{\ell}$ equals 0 (as in your provided example), but we want to emphasize that the minimal eigenvalue of $\nabla^2 \ell$, i.e., $\mu_j$ in Theorem 3.3, is not 0 since $\nabla^2 \ell$ is proven to be positive-definite in Appendix D, lines 714-738. Making uninformative alterations, as in your provided example, can be viewed as an extreme sample from $n$ potential possible alterations ($n$ in Eq. (4)).
>
> We apologize for the unclear presentation, and we will refine the related part. Thank you!
>
> ---
>
> **Q1.** Exploration-exploitation tradeoff.
>
> **A3.** Thanks for your question. Our modeling approach is different from conventional RL methods [4]. For the AUF problem, there are few opportunities to take actions (and perform explorations); hence, exploiting all available information to make effective decisions is essential. By leveraging the structural information (SRM), the rehearsal-based method can make decisions effectively without exploring different decision actions, as illustrated in the experimental results. Additionally, as explained in A2 above, selecting alterations does not adversely affect parameter estimation, eliminating the need for an exploration-exploitation tradeoff.
>
> We hope that these explanations better convey our points. Thanks again!
>
> ---
>
> **References:**
>
> [1] On implementing a primal-dual interior-point method for conic quadratic optimization, Math. Program. 2003.
>
> [2] Interior-point polynomial algorithms in convex programming, SIAM 1994.
>
> [3] Rehearsal learning for avoiding undesired future, NeurIPS 2023.
>
> [4] Reinforcement learning: An introduction, MIT Press 2018

---

> > ### Comment · Reviewer_weAS · 2024-08-11
> >
> > Thank you for your response. It addresses my concerns and I have raised the score.

---

> ### Author Response · Authors · 2024-08-11
>
> Dear Reviewer weAS,
>
> We are pleased to be able to address your concerns. Once again, thanks for the time and effort you dedicated to reviewing our work.
>
> Best Regards,
>
> Authors

---

### Official Review · Reviewer_w1Mx · 2024-07-12

**Soundness:** 4
**Presentation:** 3
**Contribution:** 4
**Rating:** 7
**Confidence:** 4

**Summary:**

In this paper, the authors address decision-making problem that sufficient interactions are not available. In this case, RL is not suitable. The authors model the structure among the observed variables, and use the structure to help the decisions. Compared to the previous studies [Qin et al. 37], the method can be used in a dynamic environment and can efficiently find the suggested decision (in polynomial time). To deal with the dynamic environment, they introduce the online learning method (Alg. 2). To efficiently find the suggested decision, they convert the optimization problem to a QCQP problem, which can be implemented in polynomial time. The experimental results verify the effectiveness.

**Strengths:**

1. The method of Qin et al. [37] suffers a high computational cost. In this paper, the authors convert the problem to a QCQP problem, which makes it computable in polynomial time. It is a valuable contribution.

2. Theorem 3.3 presents an interesting and sensible theoretical guarantee. It is novel to see that some traditional online learning methods could be used in such decision tasks.

**Weaknesses:**

Some discussion about the offline RL are missing. See Questions for the details.

Given the results of theorem 3.5: I do not know where $\tau$ is reflected in your algorithm. It seems that $\tau$ is never mentioned in Section 3.3. It is a bit wired, and needs more illustrations.

The writing could be improved. There are some weird sentences. I suggest the authors carefully revise the paper. For example, "We provide the theoretical guarantees of our method, and experimental results validate the effectiveness and efficiency of the method." -> "We provide the theoretical guarantees for our method. Experimental results validate the effectiveness and efficiency of the method."

**Questions:**

I agree that RL is not suitable for the setting. However, I am wondering why offline RL cannot be used instead? Relevant discussions are missing.

I can understand that the problem is hard in the non-linear case. Could authors have some discussions for the case that the data is non-linear?

**Limitations:**

No limitations.

---

> ### Author Rebuttal · Authors · 2024-08-06
>
> Thanks for the insightful feedback and the interest in our work! We hope our responses can address your concerns.
>
> **W1&Q1.** Discussion on the offline RL.
>
> **A1.** Thanks for your question. Generally speaking, online-offline hybrid RL methods can reduce the number of interactions by leveraging offline policy learning. However, these methods do not fit the AUF scenarios described in the paper for the following reasons:
>
> - In the offline training stage, hybrid RL methods use labeled offline datasets, i.e., offline datasets containing $(s, a, r)$ samples. However, in the AUF scenarios presented in the paper, only a few observational samples are available (in $(s, r)$ form), which do not contain information on any actions.
> - Considering "no action" as a special type of action is practical, but this approach would transform all offline data into the $(s, \text{no-action}, r)$ form, making it difficult to learn effective offline policies.
> - To achieve an effective policy, hybrid RL methods require a large number of offline samples and online interactions compared to the rehearsal-based methods [1]. For example, millions of online and offline samples are typically needed to obtain an effective policy [2, 3]. In contrast, our approach works well with only 100 samples in the AUF setting. This is because the rehearsal-based method can leverage the fine-grained structural information contained in the involved variables, whereas RL methods cannot.
>
> We will add this discussion to the revised paper. Thanks again!
>
> ---
>
> **W2.** $\tau$ in Theorem 3.5 and Algorithm 3.
>
> **A2.** We apologize for the confusion. In fact, the threshold $\tau$ is used in Eq. (6) to construct the matrix $\mathbf{P}=(\chi^{-1}(\tau)\mathbf{C}\mathbf{\Sigma}\mathbf{C}^\top)^{\frac{1}{2}}$, which appears in the constraint term in Eq. (6). Although $\tau$ is not explicitly mentioned in Section 3.3, both Theorem 3.5 and Algorithm 3 are related to the optimization Eq. (6). Hence, Theorem 3.5 and Algorithm 3 are connected to the threshold $\tau$ as well.
>
> We will clarify the related expressions in the revised version. Thanks!
>
> ---
>
> **W3.** Writing problems.
>
> **A3.** We truly appreciate your advice. We will review and refine the expressions in the paper. Thank you!
>
> ---
>
> **Q2.** Discussion for the non-linear case.
>
> **A4.** Generally speaking, non-linearity is indeed a significant challenge. However, it can be addressed in some cases, and we discuss the non-linearity in the following two stages:
>
> - For estimating the parameters of the SRM. In this case, non-linearity leads to different loss and surrogate loss functions compared to Eq. (4) and Eq. (5). If the new loss function $\ell_{\text{new}}$ is also convex w.r.t. $\beta$, then our proposed Algorithm 2 can still be used to estimate the parameters sequentially. Furthermore, Theorem 3.3 and Proposition 3.4 remain applicable.
> - For making decisions based on the estimation. When the relationships among variables are non-linear, Lemma 3.1 no longer holds. As an alternative, one can consider $\mathbf{Y}_t = f(\mathbf{x}_t, \mathbf{z}_t^\xi, \mathbf{\epsilon}_t)$, where $f$ is a non-linear function. To obtain a probability region as described in Proposition 3.2, the characteristic function [4] of the random vector may be useful. In this scenario, the constraint in Eq. (6) might no longer be linear or quadratic, which means the QCQP reduction may not apply. However, if the constraint is convex, optimization algorithms such as projection gradient descent [5] can be used to solve the new optimization problem.
>
> Lastly, we want to emphasize that addressing non-stationarity and improving time complexity are challenging even in the linear case. We will include this discussion in the future work section of the paper. Thank you!
>
> ---
>
>
> **References:**
>
> [1] Rehearsal learning for avoiding undesired future, NeurIPS 2023.
>
> [2] Hybrid RL: Using both offline and online data can make RL efficient, ICLR 2022.
>
> [3] Offline meta reinforcement learning with online self-supervision, ICML 2022.
>
> [4] Elementary probability theory, Springer 1977.
>
> [5] Convex optimization. Cambridge University Press 2004.

---

> > ### Comment · Reviewer_w1Mx · 2024-08-12
> >
> > These responses address my concerns and questions well, and thus further solidify my rating. Thanks.

---

> > > ### Author Response · Authors · 2024-08-12
> > >
> > > Dear Reviewer w1Mx,
> > >
> > > Thanks for your positive feedback. We are glad that our responses addressed your concerns and contributed to your evaluation.
> > >
> > > Best Regards,
> > >
> > > Authors

---

### Official Review · Reviewer_pTXc · 2024-07-17

**Soundness:** 4
**Presentation:** 4
**Contribution:** 4
**Rating:** 7
**Confidence:** 3

**Summary:**

The authors formulate the Avoiding Undesired Future (AUF) problem in real-world scenarios of decision-making, especially in non-stationary environments, and propose a method to avoid undesired outcomes with minimal costs. Here the non-stationarity majorly comes from the different costs corresponding to different actions, and the varying influence relations over time. They also provide theoretical guarantees of their method and empirical results demonstrate the effectiveness and efficiency of the proposal.

**Strengths:**

- This paper is written well and clearly, with intuitive motivation and clarified novelty.

- This paper includes a complete theoretical analysis and algorithmic design. Their proposed problem formalization is more general and practical than existing methods [37]. In particular, they first proposed a sequential method to maintain the dynamical influence, with guarantees of estimation error bound. They entailed Proposition 3.2 and Theorem 3.5 to help find the efficient alteration for $Z_t$ with the minimal cost. They finally propose the whole algorithm called AUF-MICNS, to avoid undesired outcomes in each decision round.

- Experimental results show the effectiveness and efficiency of their proposed algorithm, where the evaluation metrics are success frequency, estimation error, average running time, etc.

**Weaknesses:**

I think my major concerns have been settled by the Supplementary Materials. So I have no other comments about the weaknesses.

**Questions:**

For the differences between SRM in the rehearsal graph and SCM in causality:
- In the linear cases, it is easy to define the coefficients as the influences. When in the nonlinear cases, how to define the influences in the rehearsal graph? Is it the same as in causation, e.g., definitions of causal influence or causal effects?
- Can the influence in rehearsal graphs (SRM) represent the bi-directional edge information? If not, I am confused what are the differences between such bi-directional relations in rehearsal learning and causality. Though in [35], the bidirectional edges are often due to common causes between two variables, there also exist some works that use causality to represent mutually influenced relationships[1*]. A causal graph can also include cycles.
- The operators in Figure 2 seem identical to the Intervention operator in causality.

[1*] Vimaleswaran K S, Berry D J, Lu C, et al. Causal relationship between obesity and vitamin D status: bi-directional Mendelian randomization analysis of multiple cohorts[J]. PLoS medicine, 2013, 10(2): e1001383.

There are other minor typo errors:
- It seems that in Eq.(1) or Eq.(3), it is better to add $t$ as a subscript for $V_j$ and $\varepsilon_j$?
- In line 181, "ound" might be "round".

**Limitations:**

Not applicable.

---

> ### Author Rebuttal · Authors · 2024-08-06
>
> Thanks for the valuable feedback and appreciation of our work. We hope that our responses could mitigate your concerns.
>
> **Q1.** The difference between SRM and SCM.
>
> **A1.** Thank you for your insightful question. To some extent, the SRM and Rh(.) operations [1, 2] are indeed similar to their counterparts, SCM and do(.), in causality [3, 4]. However, there are some differences between the two graphical models. We will first outline these differences and then address the specific aspects you are interested in.
>
>
> 1. The modeling granularity of the rehearsal graph is more flexible. Under the assumption of causal sufficiency [3], direct edges in a DAG (directed acyclic graph) represent causal linkages. For example, $A \rightarrow C$ indicates that $A$ is the direct cause of $C$. In contrast, direct edges in a rehearsal graph do not necessarily represent causal linkages. For instance, $A \rightarrow C$ only implies that changes in $A$ lead to changes in $C$, without stating that $A$ is the direct cause of $C$. In this sense, the rehearsal graph is similar to a MAG (maximal ancestral graph) [5], however:
>
> 2. Bi-directional edges in a MAG do not have the same meaning as bi-directional edges in the SRM. Specifically, $A \leftrightarrow B$ in a MAG indicates that there are common causes between $A$ and $B$. Consequently, $do(A)$ would remove all associations between $A$ and $B$. In contrast, $A \leftrightarrow B$ in a rehearsal graph signifies that $A$ and $B$ mutually influence each other. Therefore, $Rh(A)$ only removes the influence of $B$ on $A$, resulting in $A \rightarrow B$ in the modified graph.
>
> 3. Possible dynamic influence relationships are allowed in SRM, as detailed in Appendix A.2.
>
> Based on the differences above, we would like to address your questions:
> - **Non-linear influences in rehearsal graph**. In this case, the influence $A \rightarrow B$ in the SRM is defined as the changes in $B$ when a unit change occurs in $A$. This is similar to the causal effect; however, the SRM also allows for dynamic properties, while the SCM generally assumes stationarity. Different from classic causations which typically describe the nature process that is stationary, the influence relation is more in accord with the decision process which could be dynamic. To prevent the potential misleading, we adopt influence relation in the paper.
>
> - **Information of bi-directional edge**. As explained in point 2 above, a bi-directional edge in causality (MAG) indicates a non-ancestral relationship. In contrast, a bi-directional edge in the rehearsal graph models a mutually influenced relationship. This modeling approach differs from cyclic causality [6, 7, 8], which also allows for mutually influenced $A _{\leftarrow}^{\rightarrow} B$ in the causal graph.
>     1. In cyclic causality modeling, parameters related to $A _{\leftarrow}^{\rightarrow} B$ exist in both observational and interventional situations [6].
>     2. In SRM, parameters related to edge $A \leftrightarrow B$ only appear when one variable is altered (no parameters are associated with bi-directional edges in observational situation). This modeling approach reduces the number of parameters in the observational case and is reasonable. Consider the following simple example as an illustration: when $x_1 \leftrightarrow x_2$, $x_3 \rightarrow x_1$, and $x_3 \rightarrow x_2$, using $x_1 = ax_2 + bx_3$ and $x_2 = cx_1 + dx_3$ to model the relations is equivalent to $x_1 = \alpha x_3$ and $x_2 = \gamma x_3$ with $\alpha = \frac{ad+b}{1-ac}$ and $\gamma = \frac{bc+d}{1-ac}$. The latter model only uses 2 parameters, $\alpha$ and $\gamma$. Additionally, if $x_1$ is altered, the structure becomes $x_1 \rightarrow x_2\leftarrow x_3$. In this case, the parameter $\beta_{x_1 x_2}$ associated with $x_1\rightarrow x_2$ occurs. More details about the bi-directional edges are discussed in [2].
> - **The $Rh(\cdot)$ operator in Figure 2 seems identical to intervention operator**. Conceptually, yes. However, as discussed in point 2 above, when operating on a node with bi-directional edges, these two operators will lead to different graph structures. This difference arises primarily from the distinct information contained in the bi-directional edges in rehearsal graph and MAG. To prevent the misunderstanding, we follow the existing operator $Rh(\cdot)$.
>
>
> At last, it is noteworthy that our approach can also work well when the structural information is expressed by an SCM (in linear Gaussain case), because Algorithm 2 and Proposition 3.2 also hold for SCM modeling.
>
> Thanks again for your helpful feedback. We will add this discussion and comparison with new related works in the revised version.
>
> ---
>
> **Q2.** Typo errors that subscript $t$ should be added.
>
> **A2.** Thanks for your advice. We will add the subscript $t$ for $V_j$ and $\varepsilon_j$ in Eq. (1) and (3).
>
> ---
>
> **Q3.** Typo errors that "ound" should be "round".
>
> **A3.** Thanks for your sharp observation. We will correct the spelling error.
>
>
> ---
>
> **References:**
>
> [1] Rehearsal: Learning from prediction to decision, FCS 2022.
>
> [2] Rehearsal learning for avoiding undesired future, NeurIPS 2023.
>
> [3] Causation, Prediction, and Search, MIT Press, 2000.
>
> [4] Causality: Models, Reasoning and Inference, Cambridge University Press, 2009.
>
> [5] Ancestral graph Markov models, The Annals of Statistics, 2002.
>
> [6] Learning linear cyclic causal models with latent variables, JMLR 2012.
>
> [7] Causal relationship between obesity and vitamin D status: Bi-directional Mendelian randomization analysis of multiple cohorts, PLoS medicine 2013.
>
> [8] NODAGSFlow: Nonlinear cyclic causal structure learning, AISTATS 2023.

---

### Decision · Program_Chairs · 2024-09-25

**Decision:**

Accept (poster)

**Comment:**

This paper studies an interesting setting that bridges predictions and causality. The authors pose a problem where the decision-maker can intervene to avoid undesired future outcomes, and wishes to due so with minimal cost. In contrast to prior work, the proposed algorithm can handle non-stationary environments—varying action costs and dynamic influence relations between variables over time. The main algorithm estimates the dynamic influence relations, based on which actions/decisions are taken. The reviewing team has provided helpful feedback, and I encourage the authors to revise the submission in order to maximize clarity. In particular, it is strongly recommended that they provide a candid discussion of the limitations of the proposed algorithm in the camera-ready version.